# On Survivor Stocks in the S&P 500 Stock Index

**Klaus Grobys** [1,2]

1   Finance Research Group, School of Accounting and Finance, University of Vaasa, Wolffintie 34,
    65200 Vaasa, Finland; klaus.grobys@uwasa.fi
2   Innovation and Entrepreneurship (InnoLab), University of Vaasa, Wolffintie 34, 65200 Vaasa, Finland

**Abstract:** This paper investigates the performance and characteristics of survivor stocks in the S&P 500 index. Using both in-sample and out-of-sample comparisons, survivor stocks outperformed this market index by a considerable margin. Relative to other S&P 500 index companies, survivor stocks tend to be small-value stocks that exhibit high profitability and invest conservatively. Surprisingly, survivor stocks tend to be loser stocks with negative exposure to the momentum factor. Further analyses show that the volatility of the survivor stocks portfolio is less exposed to tail risks and responds less to shocks in the innovation process.

**Keywords:** asset pricing; S&P 500 index; survivor stocks; risk factors; momentum

**JEL Classification:** G10; G12; G15; G19

## 1. Introduction

The purpose of this paper is to investigate the performance of long-run survivor stocks in the S&P 500 index and their characteristics. We make use of Standard & Poor's 2 March 2007 announcement and use the CRSP database to retrieve data for all survivor firms that exist until December 2019 but may have dropped out of the S&P 500 index in the ex post 2 March 2007 announcement period. We refer to this stock portfolio as *all survivors*. The statistical properties of this survivor portfolio are compared to the S&P 500 index. Additionally, we examine the survivor portfolios' outperformance relative to the index in general as well as risk-adjusted performance. Treating the ex ante March 2007 period as in sample and the ex post March 2007 period as out of sample, we further investigate whether a structural change occurred in the performance of survivor stocks in the ex post announcement period. In addition, we replicate our analysis using publicly available data retrieved from Yahoo. Lastly, in an effort to gain a deeper understanding of the performance of our survivor stocks portfolio, we explore the dynamics of factor exposures across time.

Our paper contributes to the academic literature in a number of ways. The S&P 500 index is widely considered to be an important gauge of the U.S. equity market and is prominently quoted in stock markets around the world (Gnabo et al. 2014). Being elected to join the constituent companies in the S&P 500 index is quite a feat: while there are more than 4000 listed companies in the U.S. stock market[1], only approximately 10% of these companies achieve membership in the S&P 500 stock index. Moreover, a company must pass the following battery of criteria to be selected by the Index Committee:[2] (1) primarily U.S. based, (2) market capitalization exceeding $8.2 billion, (3) highly liquid shares[3], (4) public trading of 50% or more of its outstanding shares, (5), positive earnings in the most recent quarter, and (6) a positive sum for the previous four quarters' earnings. Only very successful companies can fulfill these strict requirements. According to Chen and Lin (2018), member companies benefit from reductions in financial constraints, and a lower cost of equity, among other advantages.[4] Unfortunately, over time, most companies eventually do not pass these criteria and are dropped from the index. Particularly relevant to the present study, on 2 March 2007, Standard & Poor published the list of companies that have been

in the S&P 500 index since March 1957. Remarkably, only 17% of the original constituent companies survived over 50 years. These long-run survivors represent less than 2% of all listed stocks in the U.S. To our knowledge, no previous study investigates both the performance and characteristics of these exceptionally successful companies.

Long-run survivors in this well-known market index are exceptional in terms of fulfilling strict criteria for membership. Survivor companies remained profitable despite many economic shocks that periodically occurred over time. The closest published study to the present work is Siegel and Schwartz (2006), who investigated the long-term returns of the original S&P 500 constituent companies from March 1957 to December 2003. The authors found that the buy-and-hold returns of the original 500 companies outperformed the returns on the continually updated S&P 500 index used by investment professionals. Their study contradicted earlier research by McKinsey & Company's Foster and Kaplan (2001), who documented that new companies added to the S&P 500 index generated higher returns than the original companies. In their study, the performance of three portfolios was examined: (1) a survivor portfolio of 125 original companies that remained intact (except possibly for a name change) from 1957 to 2003, (2) a portfolio of direct descendants consisting of the shares of companies in the survivors portfolio plus the shares issued by companies that acquired an original S&P 500 company, and (3) a portfolio of total descendants including all companies in the direct descendants portfolio and all the spin-offs and other stocks distributed by the companies in the portfolio of direct descendants. Their results indicated that differences in average returns between these portfolios were negligible.[5] Our approach to identifying survivor stocks differs from theirs by using those companies announced as survivor companies by Standard & Poor on 2 March 2007. Additionally, we expand their sample period beyond 2003 to encompass the 2008–2010 global financial crisis. In doing so, as proposed in Harvey et al. (2016), our analyses take into account multiple testing hurdles.

Numerous studies have investigated companies in the S&P 500 index. Chan et al. (2013) explored the long-term effects of S&P 500 index additions and deletions on sample stocks from 1962 to 2003. The authors documented significant long-term price increases for both categories of stocks, with deleted stocks outperforming added stocks. As already noted, they found that firms added to the S&P 500 index gain a competitive edge in terms of reductions in financial constraints and the cost of equity. Platikanova (2016) examined revisions in cash holdings and the market valuation of investment opportunities of 475 firms added to the S&P 500 in the 1980–2010 period. They found a larger decrease in cash for index inclusions in sectors with high financial dependence. Shankar and Miller (2006) investigated market reactions to S&P 600 index inclusions and deletions. They observed significant announcement effects in terms of price increases, trading volume, and institutional ownership. Afego (2017) provided an excellent literature survey on the effects of changes in stock index composition. He argued that the vast majority of studies in this research area focused on price and volume effects for S&P 500 index firms due to the enormous value of investment assets directly benchmarked to the index. The survey revealed that S&P 500 stocks face significant short-term price pressures due to exceptionally high trading volumes by tracker funds with an estimated $2.2 trillion in directly-linked funds. Finally, Chen and Lin (2018) documented that companies in the S&P 500 index gain a competitive advantage over non-S&P 500 industry competitors in terms of positive stock valuation effects at the expense of competitors. Index inclusion is associated with both a decrease in the cost of equity and an increase in capital investment for newly added firms. Our study contributes to this literature by focusing on what we can learn from those companies that survived in the S&P 500 over a long period of time.

Interestingly, we find that the risk-adjusted average excess return of our portfolio of survivor stocks is 5.16% per annum after controlling for the excess returns of the S&P 500 index. This finding supports Siegel and Schwartz (2006), who documented that the original S&P 500 constituent stocks outperformed the index. Our findings indicate that this outperformance is even more pronounced after controlling for market risk. Relative to the S&P 500 index, we find that survivors tend to be on average small-value stocks that exhibit

high profitability and conservative capital investment. Moreover, survivor stocks' returns are negatively correlated with momentum returns, which suggests that their returns more closely mimic losers rather than winners in momentum portfolios. During the financial crisis of 2008 and 2009, survivor stocks earned higher profits, invested more aggressively in capital, and decreased in size over time relative to the S&P 500 index. Additionally, the value characteristic of survivor stocks appears to be sample specific to the post-financial crisis period. We infer that survivor companies were better able to withstand the stresses of economic downturns than other S&P 500 index firms.

Using Standard & Poor's announcement on March 2007 as a structural break, we further explore whether survivor companies thereafter experienced a decrease in performance as measured by their average excess returns until December 2019. Since the evidence does not support this pattern, our results again support Siegel and Schwartz (2006). We further investigate the volatility process of the survivor stock portfolio as opposed to the S&P 500 index. We find that the volatility of the survivor stocks portfolio responds less to shocks in the return-generating innovation process than the index. This finding is surprising given the small fraction of survivor stocks in the index. Moreover, we find that the portfolio of survivor stocks is less exposed to fat tails than the index, such that investors are less exposed to extreme events. Finally, replicating the analyses using publicly available data for survivors on Yahoo Finance, our results using CRSP data to construct the survivors portfolio are corroborated. Based on the empirical evidence, we conclude that survivor stocks are different from other stocks in the S&P 500 index with remarkable resilience to withstand economic downturns and coincident stock market collapses.

The next section describes the data. Section 3 discusses our methodological approach. Section 4 provides the empirical results. The last section concludes.

## 2. Data

On 2 March 2007, Standard & Poor, the world's leading index provider, released the list of survivor companies in the S&P 500 index from March 1957 to March 2007. The list is publicly available on the internet.[6] Interestingly, only 86 original constitute firms of this well-known market index survived over the past 50 years, which corresponds to 17.20% of the 500 original constitute firms.[7] We begin our data collection as follows: (i) use the survivor list of company names; (ii) search the corresponding stock ticker; and (iii) employ the CRSP database to match the stock ticker with corresponding stock returns.[8] For these 86 survivor companies, we identified available data associated with 92 stocks. Table A1 in the Appendix A lists the firm names. The number of stocks over time is plotted in Figure 1. Using survivor stocks, we compute an equal-weighted average portfolio denoted as the *all survivors portfolio* ($RET_{SURVIVOR}^{ALL}$).[9]

This figure illustrates the number of available survivor stock observations over time using the CRSP database. Additionally, we retrieve monthly data for the Fama and French (2018) risk factors (viz., six-factor model) and Treasury bill rate from Kenneth French's website. Since data for the profitability factor (*RMW*) and investment factor (*CMA*) are not available before July 1963, we download data series for the size (*SMB*) and value (*HML*) factors as well as *RMW* and *CMA* factors from July 1963 to November 2019. Table 1 provides descriptive statistics for portfolios $RET_{SURVIVOR}^{ALL}$, *SMB*, *HML*, *RMW*, *CMA*, and the S&P 500 index. As shown there, the average gross return of $RET_{SURVIVOR}^{ALL}$ is 40 basis points per month higher than the average gross return of the S&P 500 index. The survivor stock portfolio exhibits a monthly standard deviation of returns equal to 3.94%, which is slightly lower that of the S&P 500 index at 4.27%. Relevant to these comparisons, it is important to bear in mind that the number of stocks in $RET_{SURVIVOR}^{ALL}$ (viz., 67) is considerably lower than the S&P 500 index.

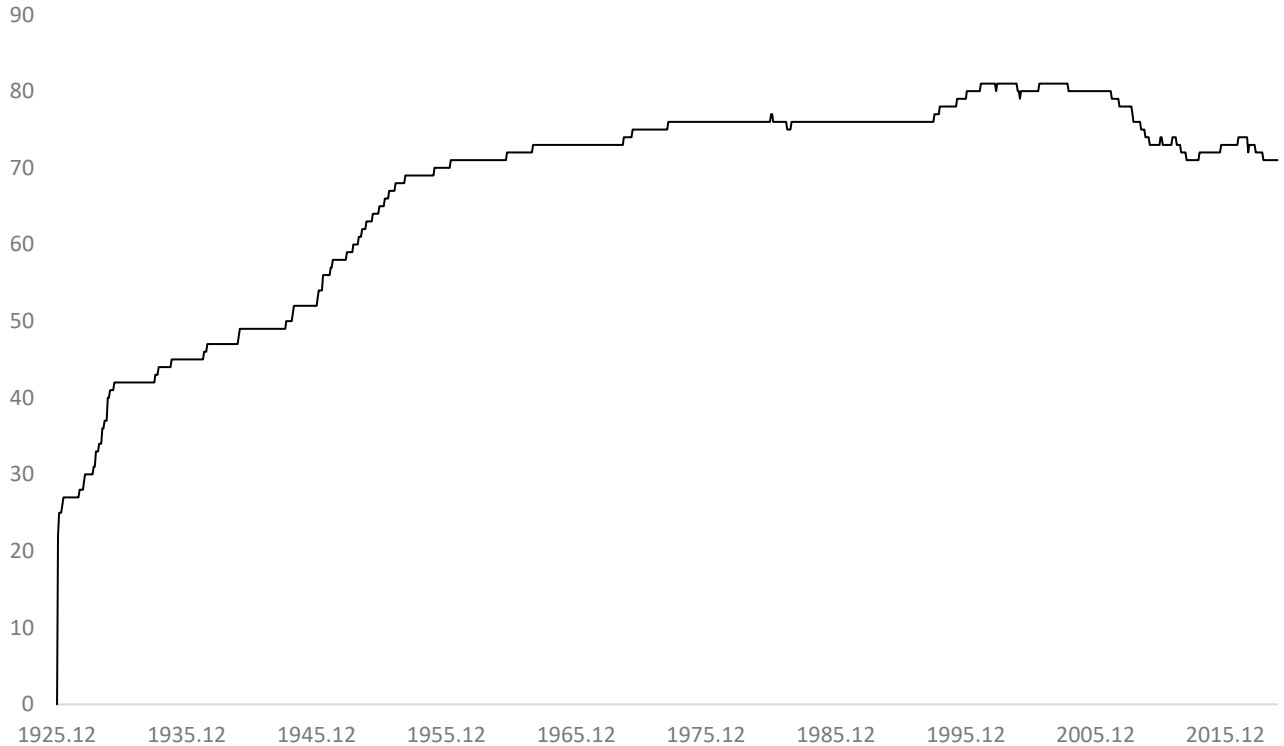

**Figure 1.** Evolution of survivor stocks in the sample period.

**Table 1.** Descriptive portfolio statistics.

| | $RET^{ALL}_{SURVIVOR}$ | S&P 500 | SMB | HML | RMW | CMA | UMD |
|---|---|---|---|---|---|---|---|
| Mean | 1.05 | 0.65 | 0.21 | 0.26 | 0.26 | 0.26 | 0.66 |
| Median | 1.28 | 0.91 | 0.09 | 0.25 | 0.22 | 0.11 | 0.71 |
| Maximum | 15.25 | 16.30 | 18.05 | 12.60 | 13.38 | 9.56 | 18.36 |
| Minimum | −18.67 | −21.76 | −14.86 | −14.11 | −18.48 | −6.86 | −34.39 |
| Std. dev. | 3.94 | 4.27 | 3.02 | 2.87 | 2.17 | 1.99 | 4.19 |
| Skewness | −0.36 | −0.44 | 0.33 | 0.01 | −0.33 | 0.32 | −1.28 |
| Kurtosis | 5.22 | 4.87 | 6.02 | 5.39 | 15.38 | 4.61 | 13.19 |

This table reports the descriptive statistics of the survivor stocks portfolio, the S&P 500 index, and the Fama and French (2018) risk factors. The sample period is from July 1963 to December 2019. The figures are given in terms of percentages.

## 3. Statistical Analysis

### 3.1. Risk Adjustments and Survivor Stock Portfolio Characteristics

Here, we investigate the outperformance of the *all survivors portfolio* relative to the S&P 500 index. For this purpose, we regress $RET^{ALL,excess}_{SURVIVOR,t}$ on the excess returns of the S&P 500 index denoted $RET^{excess}_{S\&P500,t}$ as follows:

$$RET^{ALL,excess}_{SURVIVOR,t} = \alpha + \beta \cdot RET^{excess}_{S\&P500,t} + u_t. \tag{1}$$

Table 2 reports the regression estimates, which confirm that survivor stocks outperformed the S&P 500 index by a large margin. $RET^{ALL,excess}_{SURVIVOR,t}$ generated an average return of 5.16% per annum in excess of $RET^{excess}_{S\&P500,t}$, with a *t*-statistic equal to 7.11 that is significant at any level.[10] The loading on $RET^{excess}_{S\&P500,t}$ is slightly less than unity, such that on average survivor stocks do not exhibit higher betas than the S&P 500 index.



**Table 2.** Regression estimates for all survivors using different asset pricing models.

| Alpha | S&P 500 | SMB | HML | RMW | CMA | UMD | $R^2$ |
|-------|---------|-----|-----|-----|-----|-----|-------|
| 0.43 *** | 0.86 *** | | | | | | |
| (7.11) | (60.11) | | | | | | 0.84 |
| 0.32 *** | 0.89 *** | 0.08 *** | 0.26 *** | | | | |
| (5.97) | (68.02) | (4.63) | (13.39) | | | | 0.88 |
| 0.21 *** | 0.92 *** | 0.15 *** | 0.16 *** | 0.26 *** | 0.20 *** | | |
| (4.10) | (72.51) | (8.39) | (6.61) | (10.41) | (5.36) | | 0.90 |
| 0.26 *** | 0.91 *** | 0.15 *** | 0.13 *** | 0.27 *** | 0.22 *** | −0.07 *** | |
| (5.16) | (72.56) | (8.75) | (5.09) | (11.31) | (6.14) | (−6.11) | 0.90 |

*** Statistically significant on a 1% level.

This table reports the results of regressing portfolio $RET^{ALL,excess}_{SURVIVOR,t}$ on the excess returns of the S&P 500 index as well and different asset pricing models. Ordinary *t*-statistics are reported in parentheses. The figures are given in terms of percentages. The sample period is from July 1963 to December 2019.

Can the outperformance of the survivor stocks be explained by exposures to the Fama and French (1993, 2015, 2018) risk factors? To address this question, we regress $RET^{ALL,excess}_{SURVIVOR,t}$ successively on the Fama and French (1993, 2015, 2018) three-, five-, and six-factor models defined, respectively, as follows:

$$RET^{ALL,excess}_{SURVIVOR,t} = \alpha + \beta_1 \cdot RET^{excess}_{S\&P500,t} + \beta_2 \cdot SMB_t + \beta_3 \cdot HML_t + u_t, \tag{2}$$

$$RET^{ALL,excess}_{SURVIVOR,t} = \alpha + \beta_1 \cdot RET^{excess}_{S\&P500,t} + \beta_2 \cdot SMB_t + \beta_3 \cdot HML_t \ldots + \beta_4 \cdot RMW_t + \beta_5 \cdot RMW_t + u_t \tag{3}$$

$$RET^{ALL,excess}_{SURVIVOR,t} = \alpha + \beta_1 \cdot RET^{excess}_{S\&P500,t} + \beta_2 \cdot SMB_t + \beta_3 \cdot HML_t \ldots + \beta_4 \cdot RMW_t + \beta_5 \cdot CMA_t + \beta_6 \cdot UMD_t + u_t. \tag{4}$$

The estimated regression results for these models are reported in rows two to four in Table 2. Several findings are worth noting. First, regardless of the asset pricing model, survivor stocks outperform the S&P 500 index. The economic magnitude of risk-adjusted returns, as measured by the regression intercepts, varies from 21 to 32 basis points per month with *t*-statistics between 4.10 and 5.97, indicating significance at any level. Predictably, the variation in the excess returns of the S&P 500 index explains 84% of the variation in the excess returns of the survivor stock portfolio. Controlling for various risk factors only marginally increases the *R*-squared value. Second, the positive loading on the size factor implies that the survivor stocks tend to be smaller stocks. However, this finding needs to be interpreted relative to the S&P 500 index; that is, survivor stocks are relatively smaller than the average index stock. Third, statistically significant exposures with respect to the value, profitability, and investment factors imply that survivor stocks tend to be value stocks that are profitable and invest conservatively. Fourth, and last, an unexpected finding is that the statistically significant loading on the momentum factor is negative in sign. Consequently, survivor stocks tend to have returns more correlated on average with loser than winner stocks. In view of the previously discussed S&P 500 index listing requirements, this finding is surprising.

To further investigate survivor stocks' characteristics relative to S&P 500 index companies, we employ a simultaneous equation model wherein $RET^{ALL,excess}_{SURVIVOR,t}$ and $RET^{excess}_{S\&P500,t}$ are modeled in the following system of equations:

$$RET^{ALL,excess}_{SURVIVOR,t} = \alpha_1 + \beta_{1,1} \cdot RET^{excess}_{CRSP,t} + \beta_{1,2} \cdot SMB_t + \beta_{1,3} \cdot HML_t \ldots + \beta_{1,4} \cdot RMW_t + \beta_{1,5} \cdot CMA_t + \beta_{1,6} \cdot UMD_t + u_{1,t} \tag{5}$$

$$RET^{excess}_{S\&P500,t} = \alpha_2 + \beta_{2,1} \cdot RET^{excess}_{CRSP,t} + \beta_{2,2} \cdot SMB_t + \beta_{2,3} \cdot HML_t \ldots + \beta_{2,4} \cdot RMW_t + \beta_{2,5} \cdot CMA_t + \beta_{2,6} \cdot UMD_t + u_{2,t}. \tag{6}$$

Due to the high contemporaneous correlation between $RET^{ALL,excess}_{SURVIVOR,t}$ and $RET^{excess}_{S\&P500,t}$, we use seemingly unrelated regression (SUR) to estimate system (5) and (6). If a set of equations has contemporaneous cross-equation error correlation (i.e., the error terms in the regression equations are correlated), SUR addresses this issue by using a two-step estimation procedure that explicitly models the cross-equation error correlation. Since the

correlation between $RET^{ALL,excess}_{SURVIVOR,t}$ and $RET^{excess}_{S\&P500,t}$ is manifested in $COV(u_{1,t},\ u_{2,t}) \neq 0$, SUR appears to be an adequate econometric model. In the previous analysis, we examined the average characteristics of survivor stocks relative to the underlying S&P 500 index. Using these equations, we conduct similar analyses but employ an overall market index proxied by the excess returns of the value-weighted CRSP index. The latter index is typically used in tests of the Fama and French (1993, 2015, 2018) asset pricing models.

The results are reported in Table 3. First, while the *t*-statistic associated with $\hat{\alpha}_1$ is statistically not different from zero, the *t*-statistic corresponding to $\hat{\alpha}_2$ is significantly negative at any statistical level. Hence, this evidence suggests that the outperformance of survivor stocks is driven by the underperformance of the S&P 500 index relative to the more general CRSP index. Second, the point estimator $\hat{\beta}_{2,2} = 0.16$ with a corresponding *t*-statistic of $-30.46$ confirms that stocks in the S&P 500 index tend to be large relative to those in the CRSP index. Since the *t*-statistic of $\hat{\beta}_{1,2}$ corresponding to $-0.00$ suggests that survivor stocks are not small stocks relative to the CRSP index, our evidence can only be interpreted to mean that survivor stocks are smaller relative to the average stock in the S&P 500. Third, even though the positive exposures with respect to *HML*, *RMW*, and *CMA* suggest that the average stock in the S&P 500 index tends to be a value firm that is profitable and invests conservatively, the exposures to these risk factors are very low in the range of only 0.02 to 0.06. By contrast, survivor stocks exhibit exposure with respect to these risk factors that are considerably larger in terms of their economic magnitudes with a range from 0.15 to 0.35 and *t*-statistics significant at any level.[11] Survivor stocks appear to perform better on all of these metrics. Fourth, and last, a surprising finding is that survivor stocks are, on average, considerably more exposed to loser stocks than the S&P 500. The exposure of the survivor stocks portfolio to the momentum factor is $-0.09$ as opposed to $-0.02$ for the S&P 500 index with respect to the momentum factor.

**Table 3.** Further asset pricing regression tests of all survivors.

| Dependent var. | Alpha | $CRSP^{excess}$ | SMB | HML | RMW | CMA | UMD | $R^2$ |
|---|---|---|---|---|---|---|---|---|
| $RET^{ALL,excess}_{SURVIVOR}$ | 0.03 (0.60) | 0.91 *** (74.79) | −0.00 (−0.00) | 0.15 *** (6.04) | 0.33 *** (13.98) | 0.26 *** (7.49) | −0.09 *** (−7.70) | 0.91 |
| $RET^{excess}_{S\&P500}$ | −0.25 *** (−15.81) | 1.00 *** (262.94) | −0.16 *** (−30.46) | 0.02 *** (3.08) | 0.06 *** (7.89) | 0.04 *** (3.48) | −0.02 *** (−5.16) | 0.99 |

*** Statistically significant on a 1% level.

This table reports regresses $RET^{ALL,excess}_{SURVIVOR,t}$ and $RET^{excess}_{S\&P500}$ on the excess returns of the CRSP index as well as other risk factors in Fama and French's (2018) six-factor model. Ordinary *t*-statistics are reported in parentheses. The figures are given in terms of percentages. The sample period is from July 1963 to December 2019.

### 3.2. Out-of-Sample Performance

It is important to recognize that our analysis incorporates information that the naïve investor did not know before March 2007 when Standard & Poor released the list of survivor S&P 500 index companies since March 1957. Here, we consider the out-of-sample question: What has been the performance of the survivor portfolio since March 2007? To explore whether survivors continued to outperform the S&P 500 index in the ex post announcement period, we add a binary dummy variable (denoted as $x_t$) to the regression models formulated in Equations (2)–(6) as follows:

$$RET^{ALL,excess}_{SURVIVOR,t} = \alpha + d \cdot x_t + \beta \cdot RET^{excess}_{S\&P500,t} + u_t \tag{7}$$

$$RET^{ALL,excess}_{SURVIVOR,t} = \alpha + d \cdot x_t + \beta_1 \cdot RET^{excess}_{S\&P500,t} + \beta_2 \cdot SMB_t + \beta_3 \cdot HML_t + u_t \tag{8}$$

$$RET^{ALL,excess}_{SURVIVOR,t} = \alpha + d \cdot x_t + \beta_1 \cdot RET^{excess}_{S\&P500,t} + \beta_2 \cdot SMB_t + \beta_3 \cdot HML_t \ldots + \beta_4 \cdot RMW_t + \beta_5 \cdot CMA_t + u_t \tag{9}$$

$$RET^{ALL,excess}_{SURVIVOR,t} = \alpha + d \cdot x_t + \beta_1 \cdot RET^{excess}_{S\&P500,t} + \beta_2 \cdot SMB_t + \beta_3 \cdot HML_t \ldots + \beta_4 \cdot RMW_t + \beta_5 \cdot CMA_t + \beta_6 \cdot UMD_t + u_t \tag{10}$$

where $x_t$ is a binary dummy variable with value equal to 0 in the pre-announcement period July 1963–March 2007 and 1 in the post-announcement period April 2007–November 2020. If risk-adjusted returns in the ex post March 2007 period, as measured by the sum $\alpha + d$, are statistically significantly lower, we expect that the $t$-statistic for parameter $d$ will be significantly negative. The results in Table 4 show that the parameter estimate $\hat{d}$ is negative in all model specifications but with an economic magnitude close to zero and, in most model specifications, statistically not different from zero. These findings suggest that, even in the post-announcement period, survivors continued to outperform the S&P 500 index.

**Table 4.** Out-of-sample performance of all survivors.

| Alpha | Dummy | S&P 500 | SMB | HML | RMW | CMA | UMD | $R^2$ |
|---|---|---|---|---|---|---|---|---|
| 0.50 *** | −0.32 ** | 0.86 *** | | | | | | |
| (7.33) | (−2.22) | (60.33) | | | | | | 0.84 |
| 0.35 *** | −0.12 | 0.89 *** | 0.08 *** | 0.25 *** | | | | |
| (5.67) | (−0.91) | (68.01) | (4.58) | (13.22) | | | | 0.88 |
| 0.23 *** | −0.12 | 0.92 *** | 0.15 *** | 0.16 *** | 0.26 *** | 0.20 *** | | |
| (4.08) | (−1.00) | (72.52) | (8.34) | (6.51) | (10.41) | (5.37) | | 0.90 |
| 0.30 *** | −0.19 * | 0.91 *** | 0.15 *** | 0.12 *** | 0.27 *** | 0.22 *** | −0.08 *** | |
| (5.33) | (−1.65) | (72.66) | (8.68) | (4.90) | (11.33) | (6.17) | (−6.25) | 0.90 |

* Statistically significant on a 10% level. ** Statistically significant on a 5% level. *** Statistically significant on a 1% level.

This table reports the results of regressing portfolio $RET_{SURVIVOR,t}^{ALL,excess}$ on the excess returns of the S&P 500 index as well as different asset pricing models. The regression models include a dummy variable denoted $d$ with a value of 0 in the period from July 1963 to March 2007 and a value of 1 in the period April 2007–December 2019. Ordinary $t$-statistics are reported in parentheses. The figures are given in terms of percentages. The sample period is from July 1963 to December 2019.

### 3.3. Time-Varying Factor Exposures

To better understand the risk determinants of survivor stocks' returns, we estimate the Fama and French (2018) six-factor model:

$$RET_{SURVIVOR,t}^{ALL,excess} = \alpha + \beta_1 \cdot RET_{S\&P500,t}^{excess} + \beta_2 \cdot SMB_t + \beta_3 \cdot HML_t + \beta_4 \cdot RMW_t + \beta_5 \cdot CMA_t + \beta_6 \cdot UMD_t + u_t, \quad (11)$$

where a 60 month window is used to estimate the parameter vector $\hat{\beta}_t = (\hat{\alpha}, \hat{\beta}_1, \hat{\beta}_2, \ldots, \hat{\beta}_6)$, and the estimation window is rolled forward one month at a time to the end of our sample period. This approach enables observation of trends over time in the estimated parameters.

In Figures 2–7, we report the time-varying factor exposures based on Fama and French's six-factor model from July 1968 to November 2020. From casual inspection of Figure 2, we see that the survivor portfolio's exposure to excess S&P 500 index returns is stable over time with beta close to unity. By contrast, Figure 3 shows that exposure to the size factor increases over time and exhibits noticeable volatility. We infer that, while survivor stocks in the S&P 500 index are large companies, as time passes, these companies become smaller with respect to this market index. This intertemporal pattern is consistent with Taleb (2012), who observed that " ... in spite of what is studied in business schools concerning 'economics of scale', size hurts you at times of stress; it is not a good idea to be large during difficult times." (Taleb 2012, p. 279). Even though survivor companies engaged in mergers and acquisitions over time (Siegel and Schwartz 2006), our findings suggest that survivor companies grew smaller in size relative to the S&P 500 index in general. Visual inspection of Figure 2 shows a clear linear trend of the survivor stock portfolio's exposure against the size factor, which reaches its peak in May 2007. This peak occurred shortly before the early phase of the financial crisis starting in the beginning of August 2007 with the seizure in the banking system precipitated by BNP Paribas announcing that it was ceasing activity in three hedge funds operating with U.S. mortgage debt. When stock prices collapsed in the wake of the financial crisis, the market capitalization of those

firms remained relatively stable. This pattern is implied by the sharp decrease from a positive exposure against the size factor in May 2007 until reaching its minimum with the economically largest negative exposure against the size factor in May 2012. A similar pattern can be seen after the stock market crises of 1972, 1987, and 1997. From an investment point of view, this finding suggests that survivor stocks may serve as safe havens in times of turmoil because their market capitalizations increase relative to the S&P 500.

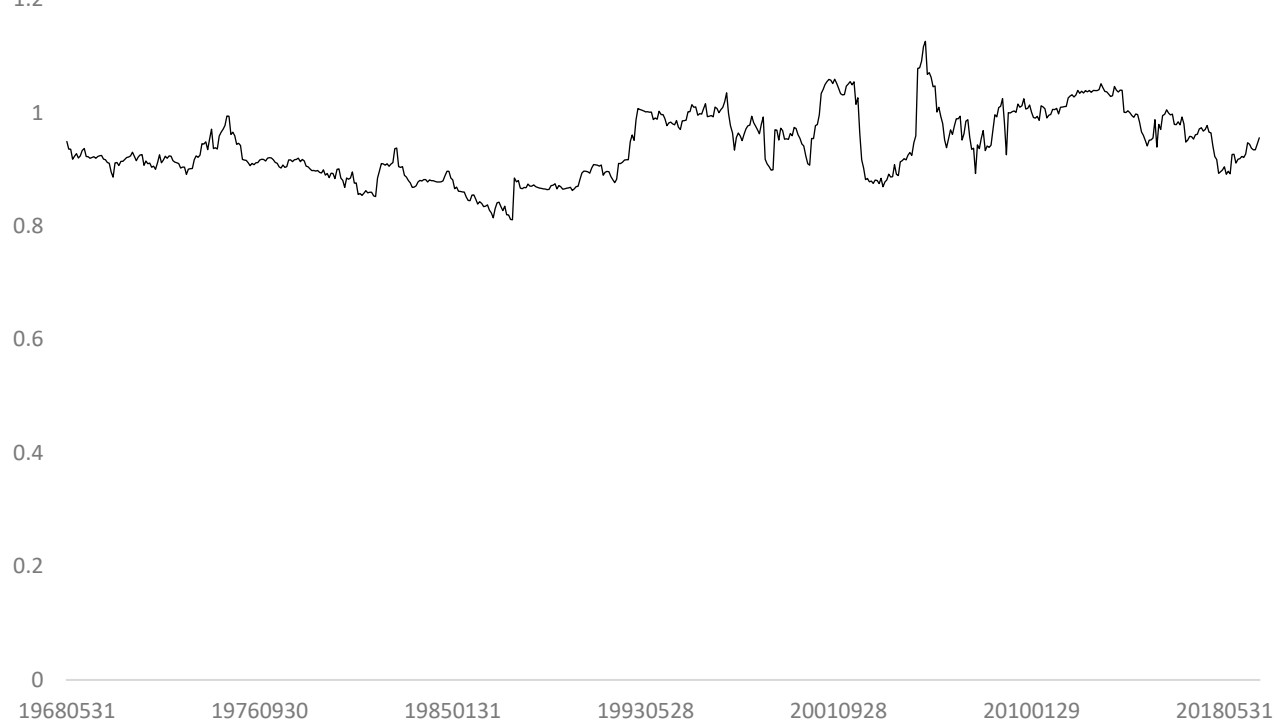

**Figure 2.** Dynamic evolution of market factor beta exposure of survivors over time.

This figure plots the dynamic evolution of excess returns of the survivor stock portfolio's time-varying market beta exposure (i.e., $\beta_1$) based on the following regression equation:

$$RET_{SURVIVOR,t}^{ALL,excess} = \alpha + \beta_1 \cdot RET_{S\&P500,t}^{excess} + \beta_2 \cdot SMB_t + \beta_3 \cdot HML_t + \beta_4 \cdot RMW_t + \beta_5 \cdot CMA_t + \beta_6 \cdot UMD_t + u_t$$

where $RET_{S\&P500,t}^{excess}$ is the excess return on the S&P 500 index, and *SMB*, *HML*, *RMW*, *CMA*, and *UMD* are the risk factors in the [Fama and French (2018)](#) six-factor model. This model is estimated iteratively on a monthly basis using a rolling time window of 60 months. The sample is from July 1968 to December 2019.

This figure plots the dynamic evolution of excess returns of the survivor stock portfolio's time-varying size beta exposure (i.e., $\beta_2$) based on the following regression equation:

$$RET_{SURVIVOR,t}^{ALL,excess} = \alpha + \beta_1 \cdot RET_{S\&P500,t}^{excess} + \beta_2 \cdot SMB_t + \beta_3 \cdot HML_t + \beta_4 \cdot RMW_t + \beta_5 \cdot CMA_t + \beta_6 \cdot UMD_t + u_t$$

where $RET_{S\&P500,t}^{excess}$ is the excess return on the S&P 500 index, and *SMB*, *HML*, *RMW*, *CMA*, and *UMD* are the risk factors of the [Fama and French (2018)](#) six-factor model. This model is estimated iteratively on a monthly basis using a rolling time window of 60 months. The sample is from July 1968 to December 2019.

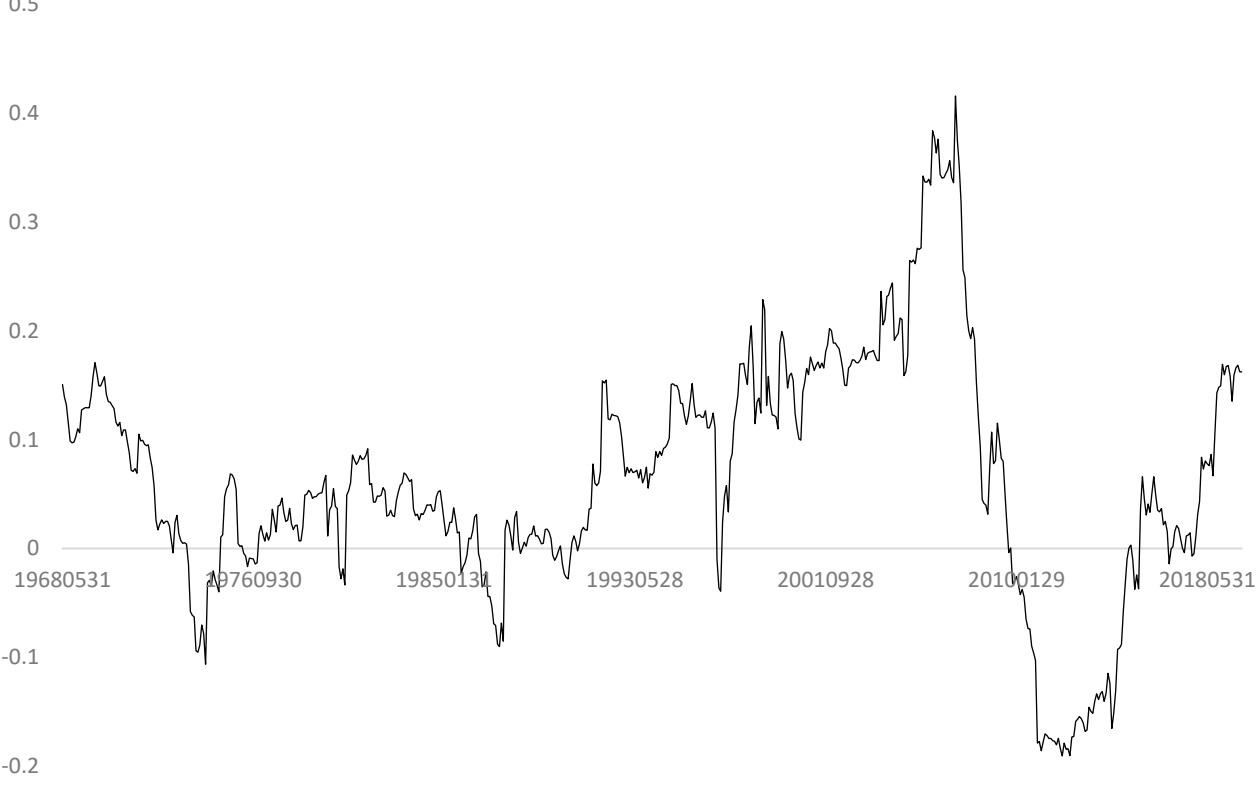

**Figure 3.** Dynamic evolution of size factor beta exposure of survivors over time.

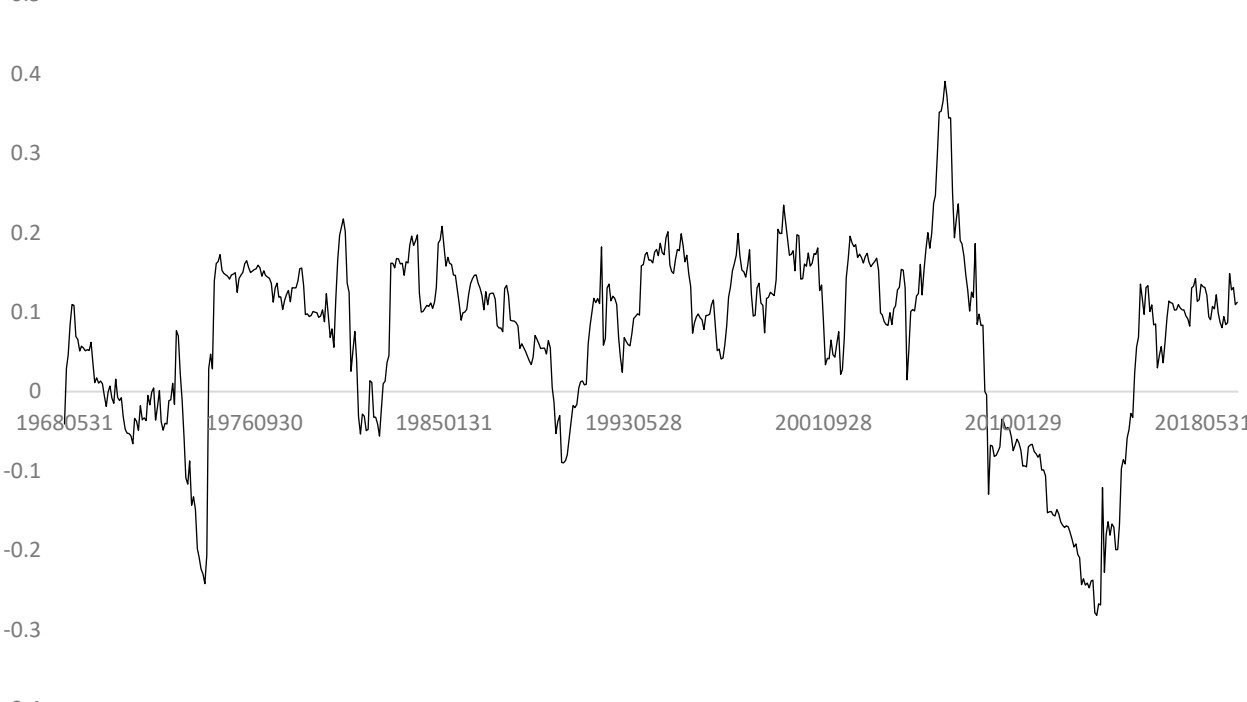

**Figure 4.** Dynamic evolution of value factor beta exposure for survivors over time.

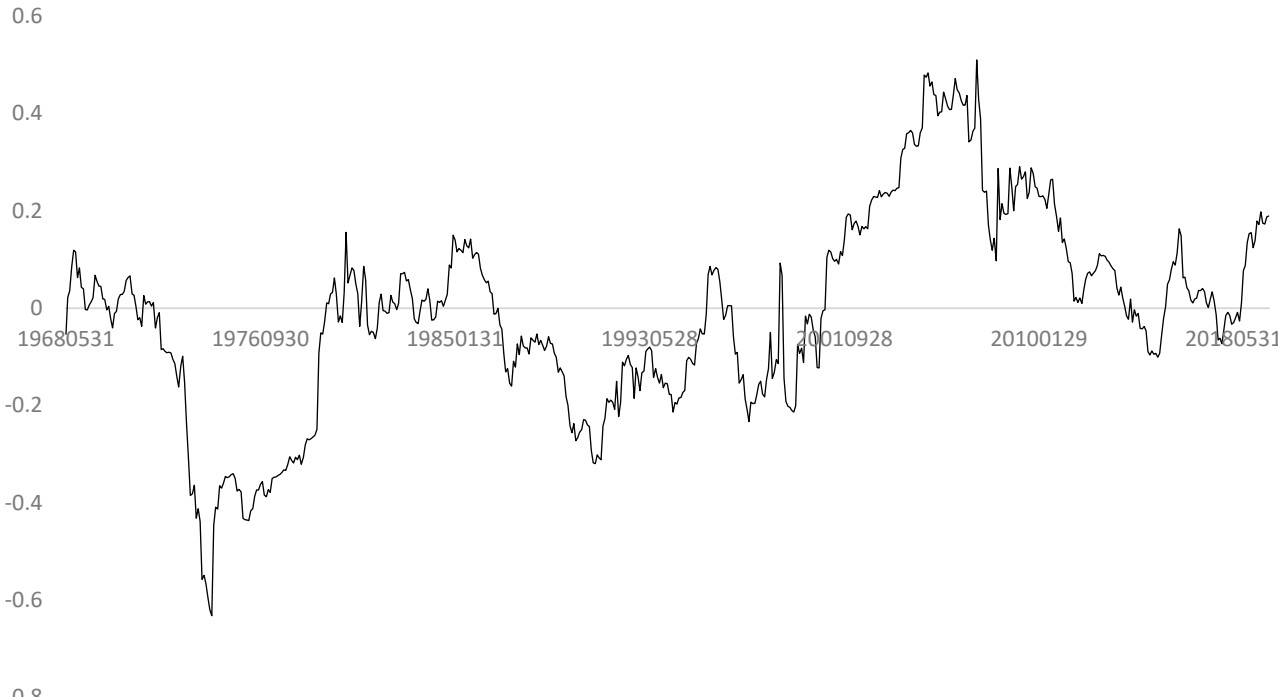

**Figure 5.** Dynamic evolution of profitability factor beta exposure for survivors over time.

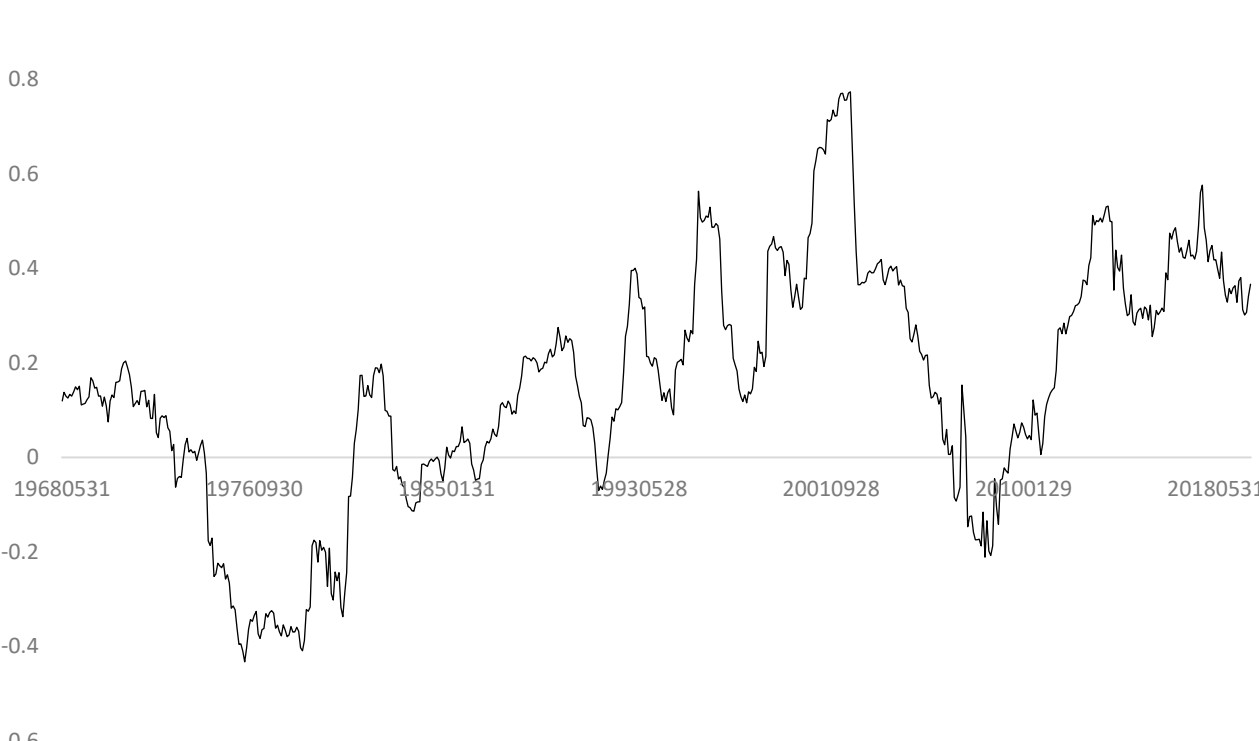

**Figure 6.** Dynamic evolution of investment factor beta exposure for survivors over time.

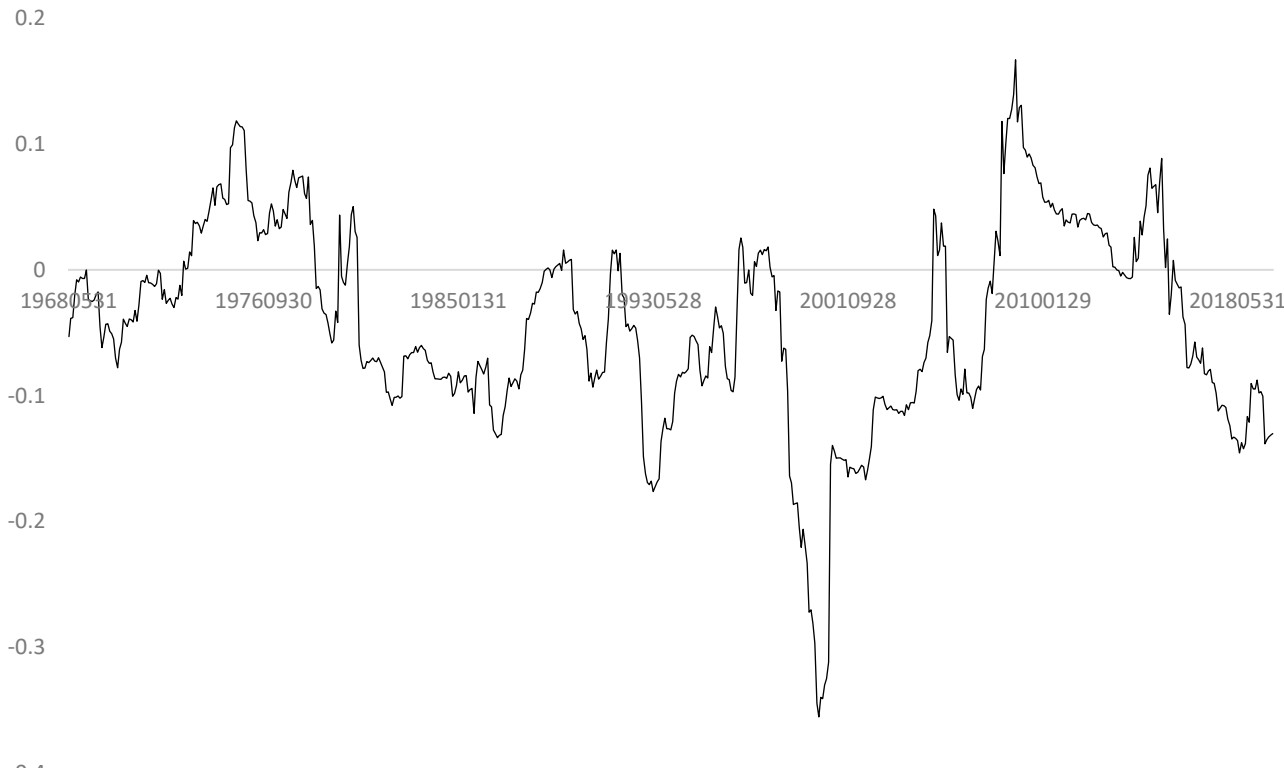

**Figure 7.** Dynamic evolution of momentum factor beta exposure for survivors over time.

Regardless of our findings in Table 2 that survivors tend to be value stocks on average, as shown in Figure 4, survivor companies experience dynamic changes in their value factor exposures. For example, from the early 1990s to 2007, survivor companies were on average value companies relative to the S&P 500 index.[12] However, in 2007 the loading on the value factor drops dramatically and thereafter continues to decrease throughout the financial crisis. From 2007 to 2015, survivor stocks were on average growth stocks. We observe a similar pattern in the mid-to-late 1980s. In sum, our findings suggest that survivor companies may be a safe haven in times of economic stress. These stocks benefit from long-term growth trends that are independent of economic cycles and tend to perform well in periods of economic downturns.

This figure plots the dynamic evolution of the excess returns of the survivor stock portfolio's time-varying value beta exposure (i.e., $\beta_3$) based on the following regression equation:

$$RET^{ALL,excess}_{SURVIVOR,t} = \alpha + \beta_1 \cdot RET^{excess}_{S\&P500,t} + \beta_2 \cdot SMB_t + \beta_3 \cdot HML_t + \beta_4 \cdot RMW_t + \beta_5 \cdot CMA_t + \beta_6 \cdot UMD_t + u_t$$

where $RET^{excess}_{S\&P500,t}$ is the excess return on the S&P 500 index, and *SMB, HML, RMW, CMA,* and *UMD* are the risk factors of the Fama and French (2018) six-factor model. This model is estimated iteratively on a monthly basis using a rolling time window of 60 months. The sample is from July 1968 to December 2019.

Extending our analyses, in Figure 5, we show that survivor companies become more profitable relative to S&P 500 index companies over time. The time-varying profitability factor exposure is on average negative until the end of the 1990s and thereafter increases and becomes positive on average.

This figure plots the dynamic evolution of excess returns of the survivor stock portfolio's time-varying profit beta exposure (i.e., $\beta_4$) based on the following regression equation:

$$RET^{ALL,excess}_{SURVIVOR,t} = \alpha + \beta_1 \cdot RET^{excess}_{S\&P500,t} + \beta_2 \cdot SMB_t + \beta_3 \cdot HML_t + \beta_4 \cdot RMW_t + \beta_5 \cdot CMA_t + \beta_6 \cdot UMD_t + u_t$$

where $RET^{excess}_{S\&P500,t}$ is the excess return on the S&P 500 index, and *SMB, HML, RMW, CMA,* and *UMD* are the risk factors of the Fama and French (2018) six-factor model. This model is

estimated iteratively on a monthly basis using a rolling time window of 60 months. The sample is from July 1968 to December 2019.

Unlike the profitability factor, the time-varying exposure against the investment factor plotted in Figure 6 exhibits a pattern similar to the time-varying value factor. While survivor stocks invested more aggressively relative to S&P 500 index companies in the beginning of the sample period, they tended to invest more conservatively relative to S&P 500 index companies in the later years. In this regard, during the financial crisis starting in 2008, the average investment factor exposure was $-0.11$ from June 2007 to June 2009; by comparison, the average exposure of the S&P 500 companies was 0.36 in this period. We infer that survivor companies were profitable firms in good financial position, which enabled them to invest more aggressively than the average company in the S&P 500 index. Companies that are profitable in times of economic stress and increase investment are attractive from the perspective of investors. While these stocks were on average value stocks in the crisis period, they became growth companies after June 2009.

This figure plots the dynamic evolution of excess returns of the survivors stock portfolio's time-varying investment beta exposure (i.e., $\beta_5$) based on the following regression equation:

$$RET^{ALL,excess}_{SURVIVOR,t} = \alpha + \beta_1 \cdot RET^{excess}_{S\&P500,t} + \beta_2 \cdot SMB_t + \beta_3 \cdot HML_t + \beta_4 \cdot RMW_t + \beta_5 \cdot CMA_t + \beta_6 \cdot UMD_t + u_t$$

where $RET^{excess}_{S\&P500,t}$ is the excess return on the S&P 500 index, and *SMB*, *HML*, *RMW*, *CMA*, and *UMD* are the risk factors of the Fama and French (2018) six-factor model. This model is estimated iteratively on a monthly basis using a rolling time window of 60 months. The sample is from July 1968 to December 2019.

Lastly, in Figure 7, we plot the time-varying momentum factor exposure of the survivor portfolio. Surprisingly, we find that, for 69% of the sample observations, the exposure against the momentum factor is negative. This negative sign in conjunction with the increasingly positive loading for profitability suggests that survivor companies are stocks that generate returns mimicking stocks with low cumulative returns in the past 12 months despite high profitability levels. This finding is interesting in view of the fact that the profitability factor is positively correlated with momentum, which implies that profitable firms tend to be winner stocks. Our findings indicate that survivor firms are the exceptions as their returns co-move with profitable loser stocks.

This figure plots the dynamic evolution of excess returns of the survivor stock portfolio's time-varying momentum beta exposure (i.e., $\beta_6$) based on the following regression equation:

$$RET^{ALL,excess}_{SURVIVOR,t} = \alpha + \beta_1 \cdot RET^{excess}_{S\&P500,t} + \beta_2 \cdot SMB_t + \beta_3 \cdot HML_t + \beta_4 \cdot RMW_t + \beta_5 \cdot CMA_t + \beta_6 \cdot UMD_t + u_t$$

where $RET^{excess}_{S\&P500,t}$ is the excess return on the S&P 500 index, and *SMB*, *HML*, *RMW*, *CMA*, and *UMD* are the risk factors of the Fama and French (2018) six-factor model. This model is estimated iteratively on a monthly basis using a rolling time window of 60 months. The sample is from July 1968 to December 2019.

### 3.4. Conditional Volatility

Does uncertainty in the survivor stocks portfolio differ from S&P 500 stocks? Since the survivor stocks portfolio contains relatively few stocks that are small compared to S&P 500 stocks, one might expect that the survivor stocks exhibit more pronounced responses to volatility shocks and are more exposed to tail risks. To explore this issue, we estimate Exponential Generalized Conditional Heteroskedasticity (EGARCH) models for both the excess returns of S&P 500 and survivor stocks as follows:

$$R^{excess}_{i,t} = \mu_i + \epsilon_{i,t},$$

$$\epsilon_{i,t} = \zeta_{i,t} \sigma_{i,t},$$

where $R_{i,t}^{excess}$ is the excess return at time $t$, $i = \{S\&P\ 500_t,\ all\ survivors_t\}$, $\mu_i$ denotes the intercept term of the mean equation, and $\epsilon_{i,t}$ is the residual term at time $t$. The equation for the variance is:

$$ln\left(\sigma_{i,t}^2\right) = c_i + \alpha_i \left|\frac{\epsilon_{i,t-1}}{\sigma_{i,t-1}}\right| + \beta_i ln\left(\sigma_{i,t-1}^2\right) + \gamma_i \frac{\epsilon_{i,t-1}}{\sigma_{i,t-1}},$$

where $\sigma_{i,t}^2$ is the conditional variance at time $t$, and the parameter vector $\boldsymbol{\theta}_{BTC} = (\mu_i, c_i, \alpha_i, \beta_i, \gamma_i)$ is estimated using maximum-likelihood estimation. As observed earlier from Table 1, given that both return series exhibit high kurtosis, we assume that the innovation process follows a fat-tailed $t$-distribution (i.e., $\zeta_{BTC,t}|\boldsymbol{\Omega}_{t-1} \sim t(v)$ with $v$ degrees of freedom).[13] Our sample period is from July 1963 to November 2020.

Table 5 reports our findings. First, using maximum-likelihood estimation accounting for fat-tailed data via the $t$-distribution supports our earlier finding—namely, the average excess returns of the survivor portfolio are economically larger than the average excess returns of the S&P 500 index. Second, the estimated alpha in the conditional variance equation of the S&P 500 index equals 0.20, which is almost twice as high as the corresponding alpha for the survivor portfolio. Both alpha estimates are statistically significant at the 5% level. This finding suggests that, despite their smaller size and numbers, the volatility of survivor stocks responds less than S&P 500 stocks to shocks in the data generating innovation process. Third, beta and gamma estimates in the variance equations are very close to each other for both portfolios, which suggests that both volatilities respond similarly to bad news in the data generating innovation process and to the long-run conditional variance. Fourth, we observe that portfolio returns of survivor stocks are less exposed to fat tails than other S&P 500 stocks, as the estimated degrees of freedom for Student's $t$-distribution is higher for the former portfolio. We infer that survivor stocks are less exposed to extreme events.

**Table 5.** Estimating volatility processes.

|  | $\mu$ | $c$ | $\alpha$ | $\beta$ | $\gamma$ | $v$ |
|---|---|---|---|---|---|---|
| S&P 500 | 0.39 *** | 0.11 | 0.20 *** | 0.90 *** | −0.16 *** | 9.53 |
|  | (2.77) | (1.28) | (2.97) | (28.26) | (−3.99) |  |
| All survivors | 0.69 *** | 0.13 * | 0.11 ** | 0.92 *** | −0.19 *** | 10.95 |
|  | (5.23) | (1.92) | (2.09) | (36.31) | (−5.39) |  |

* Statistically significant on a 10% level. ** Statistically significant on a 5% level. *** Statistically significant on a 1% level.

This table reports the estimates for the EGARCH model with mean equation:

$$R_{i,t}^{excess} = \mu_i + \epsilon_{i,t}$$

$$\epsilon_{i,t} = \zeta_{i,t}\sigma_{i,t},$$

where $R_{i,t}^{excess}$ is the excess return of at time $t$, $i = \{S\&P\ 500_t,\ all\ survivors_t\}$, $\mu_i$ denotes the intercept term of the mean equation, and $\epsilon_{i,t}$ is the residual term at time $t$. The equation for the variance is:

$$ln(\sigma_{i,t}^2) = c_i + \alpha_i \left|\frac{\epsilon_{i,t-1}}{\sigma_{i,t-1}}\right| + \beta_i ln\left(\sigma_{i,t-1}^2\right) + \gamma_i \frac{\epsilon_{i,t-1}}{\sigma_{i,t-1}},$$

where $\sigma_{i,t}^2$ is the conditional variance at time $t$, and the parameter vector $\boldsymbol{\theta}_{BTC} = (\mu_i, c_i, \alpha_i, \beta_i, \gamma_i)$ is estimated using maximum-likelihood estimation. The models assume that the innovation process follows a fat-tailed $t$-distribution (i.e., $\zeta_{BTC,t}|\boldsymbol{\Omega}_{t-1} \sim t(v)$ with $v$ degrees of freedom). The $z$-statistics are given in parentheses. The sample period is from July 1963 to December 2019.

## 4. Robustness Checks

### 4.1. The Multiple Testing Problem

Using a multiple testing framework to derive threshold levels for testing statistical significance, Harvey et al. (2016) re-evaluated 296 cross-sectional asset pricing phenomena. Their findings showed that 27% to 53% are likely false discoveries. Following these authors, we use the higher cut-off corresponding to 3.39 for testing statistical significance. Our main results in Tables 2–4 remain statistically significant. For instance, irrespective of which asset pricing model is used for risk adjusting the survivor portfolio, the regression intercepts reported in Table 2 exceed 3.39 by a large margin. In this respect, the lowest *t*-statistic of 4.10 is generated when using the Fama and French (2015) five-factor model.[14]

### 4.2. Replication Using Publicly Available Data

Hou et al. (2020), who conducted an extensive replication of 452 asset pricing anomalies, found that approximately 80% of these anomalies fail scientific replication. Subsequently, the authors recommended scientific replications of test results. To address this issue, we replicate our analyses using publicly available data from Yahoo.[15] Matching the data from Standard & Poor's announcement on 2 March 2007 with the database provided by Yahoo, we find data available for 71 stocks. Among these stocks, we excluded Raytheon Technologies Corporation (RTX) due to extreme outliers in the sample period.[16] Descriptive statistics for the final sample of 70 survivor stocks are shown in Appendix A Table A2. In Appendix A Figure A1, we plot the evolution of available survivor stocks over time (i.e., survivors are added as Yahoo Finance stock data becomes available from the earliest date of February 1962). Among these stocks, 14 survivor stocks had complete return series available from February 1962 to November 2020.[17] The final sample of 70 observations are used to form the replicated *all survivors portfolio* (denoted $RET_{SURVIVOR}^{ALL}$) which is equal weighted. As before, we retrieved data for the Fama and French (2018) risk factors (viz., six-factor model) and Treasury bill rate from Kenneth French's website. Since data for the profitability factor (*RMW*) and investment factor (*CMA*) are not available before July 1963, we retrieve data for the size factor (*SMB*), value factor (*HML*), *RMW*, and *CMA* from July 1963 to December 2019. Descriptive statistics for portfolio $RET_{SURVIVOR}^{ALL}$, *SMB*, *HML*, *RMW*, *CMA*, and the S&P 500 index are provided in Table A3. Next, we run the same regressions as in Equations (1)–(4). The results are reported in Table A4. Again survivor stocks outperformed the S&P 500 index by a considerable margin. $RET_{SURVIVOR,t}^{ALL,excess}$ generated an average return of 5.88% per annum in excess of $RET_{S\&P500,t}^{excess}$, with a *t*-statistic equal to 7.01 that is significant at any level.

The loading on $RET_{S\&P500,t}^{excess}$ is slightly less than unity, such that on average survivor stocks do not exhibit higher betas than the S&P 500 index. Employing different variations of the Fama and French (1993, 2015, 2018) does not change our results. Here, the economic magnitudes of risk-adjusted returns, as measured by the regression intercepts, varies from 27 to 40 basis points per month with *t*-statistics between 4.50 and 6.31 indicating statistical significance at any level. Note that variation in the excess returns of the S&P 500 index explains 83% of the variation in the excess returns of the survivor stock portfolio. Controlling for various risk factors negligibly increases the *R*-squared value. Second, the positive loading on the size factor implies that the survivor stocks tend to be smaller stocks. Third, statistically significant exposures with respect to the value, profitability, and investment factors imply that our replicated portfolio of survivor stocks is, on average, exposed to value stocks that are profitable and invest conservatively. Finally, we find a statistically significantly negative loading on the momentum factor. As a consequence, survivor stocks tend to have returns more correlated on average with loser than winner stocks. In sum, our results strongly confirm the key findings of our previous analysis based on CRSP data.

### 4.3. Survivor Stocks' Characteristics Relative to S&P 500 Index

Next, to further investigate survivor stocks' characteristics relative to S&P 500 index companies, we employ the simultaneous equation model in Equations (5) and (6). Based on SUR econometric estimation, the results are reported in Table A5. First, while the *t*-statistic associated with $\hat{\alpha}_1$ is statistically not different from zero, and the *t*-statistic corresponding to $\hat{\alpha}_2$ is significantly negative at any statistical level. This result implies that the outperformance of survivor stocks is driven by the underperformance of the S&P 500 index relative to the more general CRSP index. Second, the *t*-statistic of $\hat{\beta}_{1,2}$ equal to $-1.81$ suggests that survivor stocks are not small relative to the CRSP index, implying that survivor stocks are smaller relative to the average stock in the S&P 500. Third, survivor stocks exhibit exposures with respect to *HML*, *RMW*, and *CMA* that are considerably larger than the ones of the S&P 500's in terms of their economic magnitudes with a range from 0.15 to 0.35 with *t*-statistics significant at any level. We infer survivor stocks appear to perform better on all of these metrics. Fourth, and last, survivor stocks are, on average, considerably more exposed to loser stocks than the S&P 500. The exposure of the survivor stocks portfolio to the momentum factor is $-0.12$ as opposed to $-0.02$ for the S&P 500 index with respect to the momentum factor. Again, our replicated portfolio of survivor stocks strongly supports the key results of our main analysis.

Further, we address the question: What has been the performance of the survivor portfolio since March 2007? To investigate if our replicated portfolio of survivors continued to outperform the S&P 500 index in the ex post announcement period, we again employ Equations (7) to (10) using our replicated survivor portfolio drawn from the Yahoo database. The results are reported in Table A6. Once again, the parameter estimate $\hat{d}$ is negative in all model specifications but with an economic magnitude close to zero and statistically not different from zero. These findings suggest that, even in the post-announcement period, our replicated portfolio of survivors continued to outperform the S&P 500 index. Finally, we explore the conditional volatility of our replicated portfolio of survivor stocks. The results, as reported in Table A7, clearly support earlier evidence in Section 3.4.

### 4.4. Equal-Weighted Market Factor

As mentioned earlier, we have valid reasons to use equal-weighted portfolios in our current study. Value weighting would distort the overall portfolio return distribution because market capitalization as a financial variable is pareto distributed, implying that if value-weighted portfolios were used, a very small number of stocks would receive extraordinarily high weights. In our study, we are interested in the revealing potential common links among survivor stocks. A valid question that may arise is, however, could the outperformance of survivor stocks be an artefact of using equal-weighted stocks in the portfolio? To explore this issue, we download 49 equal-weighted industrial portfolios from Kenneth French website, compute the simple average return and subtract the U.S. risk free rate.[18] We use this portfolio as proxy for an equal-weighted U.S. market factor in excess form. Again, we make use of a multiple equation model as in Section 3, that is, we estimate

$$RET_{SURVIVOR,t}^{ALL,excess} = \alpha_1 + \beta_{1,1} \cdot RET_{EQUAL,t}^{excess} + \beta_{1,2} \cdot SMB_t + \beta_{1,3} \cdot HML_t + \beta_{1,4} \cdot RMW_t + \beta_{1,5} \cdot CMA_t + \beta_{1,6} \cdot UMD_t + u_{1,t} \quad (12)$$

$$RET_{S\&P500,t}^{excess} = \alpha_2 + \beta_{2,1} \cdot RET_{EQUAL,t}^{excess} + \beta_{2,2} \cdot SMB_t + \beta_{2,3} \cdot HML_t + \beta_{2,4} \cdot RMW_t + \beta_{2,5} \cdot CMA_t + \beta_{2,6} \cdot UMD_t + u_{2,t} \quad (13)$$

where $RET_{EQUAL,t}^{excess}$ is our proxy for an equal-weighted U.S. market factor in excess form and all other notation is as before. The results are reported in Table A8. We observe from Table A8. that the survivor stocks portfolio generates a risk-adjusted payoff of 27 basis points per month, whereas the S&P 500 underperforms the equal-weighted portfolio by 28 basis points per month. Testing the parameter difference $(\hat{\alpha}_1 - \hat{\alpha}_2) = 0.56$ for statistical significance, gives us a value of 57.03 for the estimated test statistic. Since the test statistic is under the null hypothesis distributed as chi-square with one degree of freedom with corresponding critical value of 3.84 for a 5% significance level, we can reject the null hypotheses (*p*-value 0.0000). Hence, the outperformance of the survivor stock portfolio

is not driven by equal-weighting the stocks in the survivor portfolio. Another interesting issue which we observe from Table A8 is that the loading against the size factor is less negative for the survivor stocks portfolio even after controlling for the equal-weighted excess market factor. Next, testing the parameter difference $(\hat{\beta}_{1,6} - \hat{\beta}_{2,6}) = -0.10$ for statistical significance gives us a value of 33.12 for the estimated test statistic. Since the test statistic is under the null hypothesis distributed as chi-square with one degree of freedom with corresponding critical value of 3.84 for a 5% significance level, we can reject the null hypotheses (*p*-value 0.0000). Hence, the survivor stock portfolio is relatively less exposed to winner stocks than the average S&P 500 firm which confirms our earlier findings.

*4.5. Additional Robustness Checks*

In our analysis, we followed the mainstream literature in using ordinary *t*-statistics (e.g., Fama and French 2015, 2017, 2018). One may wonder whether our results hold when accounting for heteroskedasticity and autocorrelation consistent (HAC) *t*-statistics.[19] To address this issue, we employ the HAC covariance matrix estimator proposed from Newey and West (1987) accounting for a lag order of $l = 1$ and replicate the main results from Table 4. The results are reported in Table A9. We observe from Table A9 that the results do not change. Hence, we infer that our results are robust with respect to potential autocorrelation and heteroskedasticity in the data.

**5. Conclusions**

On 2 March 2007, Standard & Poor released a list of companies that had been in the S&P 500 index since March 1957. Over this 50 year period, only 86 companies survived index membership requirements. Companies listed in the S&P 500 index are special in the sense that they are leading companies influential to the U.S. and global economies. A number advantages accrue to members, including reductions in financial constraints, the cost of equity, and other shadow costs, among others. Due to relatively high hurdles for membership, most companies drop out of the index over time. This study sought to investigate the performance and characteristics of survivor stocks in the S&P 500 index. Due to data availability, our survivor stocks covered the period from July 1963 to December 2019.

We found that survivor stocks outperformed the S&P 500 index by a large margin in this sample period. Their outperformance was unchanged after taking into account checks revealed that this phenomenon is not sample period specific. Relative to S&P 500 companies, survivor stocks tend to be, on average, small-value stocks that exhibit high profitability and invest conservatively. A surprising finding was that survivor stocks also tend to be loser stocks with negative exposure to the momentum factor. Further analyses revealed that survivor stocks decreased in size over time relative to other S&P 500 companies. Additionally, the value characteristic of survivor stocks shifted to be consistent with growth in periods of economic distress. Unlike other index stocks, survivors were relatively profitable and increased capital investments in times of economic stress. In this regard, survivors' returns were less exposed to fat tails than other S&P 500 stocks. Further analyses revealed that the survivor stock portfolio outperformed the S&P 500 index even in the post-March 2007 period after the public announcement by Standard & Poor's list of 50 year survivor companies. Additionally, replicating the survivor portfolio using publicly available data corroborated our findings. We conclude that survivor stocks are different from other stocks in the S&P 500 index, with remarkable resilience to withstand economic downturns and coincident stock market collapses. Comparative research is recommended on survivor stocks in other major stock markets around the world. Are survivor characteristics local or global in nature? Moreover, future research is encouraged to explore the return evolution for firms exiting the S&P 500. Since this is beyond the scope of this paper, this issue is left for future research.

**Funding:** This research received no external funding.

**Data Availability Statement:** Not applicable.

**Conflicts of Interest:** The authors declare no conflict of interest.

**Appendix A**

**Figure A1.** Evolution of survivor stocks in the sample period.

This figure illustrates the number of available survivor stock observations over time using the Yahoo database.

**Table A1.** Survivor firms based on the CRSP database.

| | | | |
|---|---|---|---|
| 1 | AMERICAN WATER WORKS & ELEC INC | 56 | AMERICAN TYPE FOUNDERS INC |
| 2 | WEST PENN ELECTRIC CO | 57 | ATF INC |
| 3 | ALLEGHENY POWER SYSTEMS INC | 58 | DAYSTROM INC |
| 4 | ALLEGHENY ENERGY INC | 59 | SCHLUMBERGER LTD |
| 5 | ALLIED CHEMICAL & DYE CORP | 60 | STANDARD OIL CO CALIFORNIA |
| 6 | ALLIED CHEMICAL CORP | 61 | CHEVRON CORP |
| 7 | ALLIED CORP | 62 | CHEVRONTEXACO CORP |
| 8 | ALLIED SIGNAL INC | 63 | CHEVRON CORP NEW |
| 9 | HONEYWELL INTERNATIONAL INC | 64 | UNION TANK CAR CO |
| 10 | ARCHER DANIELS MIDLAND CO | 65 | TRANS UNION CORP |
| 11 | BURROUGHS ADDING MACH CO | 66 | UNITED STATES STEEL CORP |
| 12 | BURROUGHS CORP | 67 | USX CORP |
| 13 | UNISYS CORP | 68 | U S X MARATHON GROUP |
| 14 | COCA COLA CO | 69 | MARATHON OIL CORP |
| 15 | CONSOLIDATED GAS CO NY | 70 | KRAFT HEINZ CO |
| 16 | CONSOLIDATED EDISON CO NY INC | 71 | WRIGLEY WILLIAM JR CO |
| 17 | CONSOLIDATED EDISON INC | 72 | AMERICAN HOME PRODUCTS CORP |
| 18 | DETROIT EDISON CO | 73 | WYETH |
| 19 | D T E ENERGY CO | 74 | SOUTHERN CALIFORNIA EDISON CO |
| 20 | DU PONT E I DE NEMOURS & CO | 75 | SCE CORP |
| 21 | EATON AXLE & SPRING CO | 76 | EDISON INTERNATIONAL |
| 22 | EATON MFG CO | 77 | ALCOA CORP |
| 23 | EATON YALE & TOWNE INC | 78 | GOODYEAR TIRE & RUBBER CO |
| 24 | EATON CORP | 79 | HERSHEY CHOCOLATE CORP |
| 25 | EATON CORP PLC | 80 | HERSHEY FOODS CORP |
| 26 | STANDARD OIL CO N J | 81 | HERSHEY CO |
| 27 | EXXON CORP | 82 | KROGER GROCERY & BAKING CO |
| 28 | EXXON MOBIL CORP | 83 | KROGER COMPANY |
| 29 | ELECTRIC BOAT CO | 84 | DOWDUPONT INC |

**Table A1.** *Cont.*

| | | | |
|---|---|---|---|
| 30 | GENERAL DYNAMICS CORP | 85 | DUPONT DE NEMOURS INC |
| 31 | GENERAL ELECTRIC CO | 86 | MELVILLE SHOE CORP |
| 32 | GENERAL MOTORS CORP | 87 | MELVILLE CORP |
| 33 | GENERAL MOTORS CO | 88 | CVS CORP |
| 34 | INGERSOLL RAND CO | 89 | CVS CAREMARK CORP |
| 35 | INGERSOLL RAND CO LTD | 90 | CVS HEALTH CORP |
| 36 | INGERSOLL RAND PLC | 91 | GENERAL MILLS INC |
| 37 | TRANE TECHNOLOGIES PLC | 92 | MCGRAW HILL PUBLISHING INC |
| 38 | INTERNATIONAL BUSINESS MACHS COR | 93 | MCGRAW HILL INC |
| 39 | FORTUNE BRANDS HOME & SECUR INC | 94 | MCGRAW HILL COS INC |
| 40 | TRANSCONTINENTAL OIL CO | 95 | MCGRAW HILL FINANCIAL INC |
| 41 | OHIO OIL CO | 96 | S&P GLOBAL INC |
| 42 | MARATHON OIL CO | 97 | KIMBERLY CLARK CORP |
| 43 | PACIFIC GAS & ELEC CO | 98 | PHELPS DODGE CORP |
| 44 | PG & E CORP | 99 | HERCULES POWDER CO |
| 45 | LOFT INC | 100 | HERCULES INC |
| 46 | PEPSI COLA CO | 101 | MINNEAPOLIS HONEYWELL REGULATOR |
| 47 | PEPSICO INC | 102 | HONEYWELL INC |
| 48 | PHILIP MORRIS & CO LTD | 103 | PENNEY J C INC |
| 49 | PHILIP MORRIS INC | 104 | PENNEY J C CO INC |
| 50 | PHILIP MORRIS COS INC | 105 | COMMONWEALTH & SOUTHERN CORP |
| 51 | ALTRIA GROUP INC | 106 | SOUTHERN CO |
| 52 | PHILLIPS PETROLEUM CO | 107 | CATERPILLAR TRACTOR INC |
| 53 | CONOCOPHILLIPS | 108 | CATERPILLAR INC |
| 54 | EASTMAN KODAK CO | 109 | COLGATE PALMOLIVE PEET CO |
| 55 | AMERICAN TYPE FOUNDERS CO | 110 | COLGATE PALMOLIVE CO |
| 111 | DEERE & CO IL | 168 | PITNEY BOWES INC |
| 112 | DEERE & CO DEL | 169 | TEXAS UTILITIES CO |
| 113 | DEERE & CO | 170 | TXU CORP |
| 114 | BRISTOL MYERS CO | 171 | ALUMINUM COMPANY AMER |
| 115 | BRISTOL MYERS SQUIBB CO | 172 | ALCOA INC |
| 116 | BOEING AIRPLANE CO | 173 | ARCONIC INC |
| 117 | BOEING CO | 174 | HOWMET AEROSPACE INC |
| 118 | ABBOTT LABS | 175 | NORTHROP AIRCRAFT INC |
| 119 | ABBOTT LABORATORIES | 176 | NORTHROP CORP |
| 120 | DOW CHEMICAL CO | 177 | NORTHROP GRUMMAN CORP |
| 121 | LOCKHEED AIRCRAFT CORP | 178 | RAYTHEON MANUFACTURING CO |
| 122 | LOCKHEED CORP | 179 | RAYTHEON CO |
| 123 | LOCKHEED MARTIN CORP | 180 | CAMPBELL SOUP CO |
| 124 | WEST VA PULP & PAPER CO | 181 | FORD MOTOR CO |
| 125 | WESTVACO CORP | 182 | FORD MOTOR CO DEL |
| 126 | MEADWESTVACO CORP | 183 | COOPER TIRE & RUBBER CO |
| 127 | WESTROCK CO | 184 | OCCIDENTAL PETROLEUM CORP |
| 128 | INTERNATIONAL PAPER & PWR CO | 185 | UNION PACIFIC CORP |
| 129 | INTERNATIONAL PAPER CO | 186 | BURLINGTON NORTHERN INC |
| 130 | PHILADELPHIA ELECTRIC CO | 187 | BURLINGTON NORTHERN SANTA FE CP |
| 131 | P E C O ENERGY CO | 188 | SEALED AIR CORP |
| 132 | EXELON CORP | 189 | CSX CORP |
| 133 | PFIZER CHAS & CO INC | 190 | NORFOLK SOUTHERN CORP |
| 134 | PFIZER INC | 191 | ALLSTATE CORP |
| 135 | COOPER BESSEMER CORP | 192 | SANTA FE FINANCIAL CORP |
| 136 | COOPER INDUSTRIES INC | 193 | NGC CORP |
| 137 | COOPER INDUSTRIES LTD | 194 | DYNEGY INC |
| 138 | COOPER INDUSTRIES PLC | 195 | ITT HARTFORD GROUP INC |
| 139 | PITTSBURGH PLATE GLASS CO | 196 | HARTFORD FINANCIAL SVCS GRP INC |
| 140 | P P G INDUSTRIES INC | 197 | QUEST DIAGNOSTICS INC |
| 141 | MINNESOTA MINING & MFG CO | 198 | SEALED AIR CORP NEW |
| 142 | 3M CO | 199 | ROCKWELL COLLINS INC |

**Table A1.** *Cont.*

| | | | | |
|---|---|---|---|---|
| 143 | MERCK & CO INC | | 200 | DYNEGY INC NEW |
| 144 | MERCK & CO INC NEW | | 201 | DYNEGY INC DEL |
| 145 | GALVIN MANUFACTURING CO | | 202 | DYNEGY INC NEW DEL |
| 146 | MOTOROLA INC | | | |
| 147 | MOTOROLA SOLUTIONS INC | | | |
| 148 | CAROLINA POWER & LIGHT CO | | | |
| 149 | CP & L ENERGY INC | | | |
| 150 | PROGRESS ENERGY INC | | | |
| 151 | CONSUMERS PWR CO | | | |
| 152 | CONSUMERS POWER CO | | | |
| 153 | C M S ENERGY CORP | | | |
| 154 | PUBLIC SERVICE ELECTRIC & GAS CO | | | |
| 155 | PUBLIC SERVICE ENTERPRISE GP INC | | | |
| 156 | HALLIBURTON OIL WELL CEMENTING | | | |
| 157 | HALLIBURTON COMPANY | | | |
| 158 | NORTHERN STATES POWER CO MN | | | |
| 159 | XCEL ENERGY INC | | | |
| 160 | MIDDLE SOUTH UTILITIES INC | | | |
| 161 | ENTERGY CORP | | | |
| 162 | ENTERGY CORP NEW | | | |
| 163 | AMERICAN GAS & ELECTRIC CO | | | |
| 164 | AMERICAN ELECTRIC POWER CO INC | | | |
| 165 | CONSOLIDATED GAS ELEC LT & PWR | | | |
| 166 | BALTIMORE GAS & ELECTRIC CO | | | |
| 167 | CONSTELLATION ENERGY GROUP INC | | | |

This table reports the firms of the corresponding survivor stocks.

**Table A2.** Descriptive statistics for survivor stocks.

| Ticker/Metric | AA | ABT | ADM | AEP | ALL | ATI | BA | BMY | BURL | CAT |
|---|---|---|---|---|---|---|---|---|---|---|
| Mean | 0.90 | 1.47 | 1.09 | 0.92 | 1.11 | 1.29 | 1.44 | 1.15 | 2.97 | 1.36 |
| Median | 0.73 | 1.36 | 1.09 | 1.10 | 1.05 | 0.36 | 1.45 | 1.11 | 3.63 | 1.17 |
| Maximum | 54.02 | 22.12 | 32.08 | 28.70 | 30.97 | 62.50 | 48.44 | 43.72 | 25.24 | 40.14 |
| Minimum | −55.59 | −20.74 | −27.36 | −17.77 | −42.78 | −50.26 | −45.47 | −28.87 | −26.73 | −35.91 |
| Std. dev. | 9.98 | 5.97 | 7.89 | 5.73 | 7.59 | 16.38 | 9.59 | 6.97 | 8.51 | 8.40 |
| Skewness | −0.02 | −0.15 | 0.12 | 0.03 | −0.72 | 0.60 | 0.23 | 0.08 | −0.49 | 0.03 |
| Kurtosis | 7.32 | 3.65 | 3.98 | 4.09 | 8.28 | 4.71 | 4.71 | 5.60 | 5.86 | 4.65 |
| Sample start | 1962-02 | 1980-04 | 1980-04 | 1970-02 | 1993-07 | 1999-12 | 1962-02 | 1972-07 | 2013-11 | 1962-02 |

| Ticker/Metric | CL | CMS | COO | COP | CPB | CSX | CVS | CVX | DD | DE |
|---|---|---|---|---|---|---|---|---|---|---|
| Mean | 1.15 | 0.65 | 1.46 | 1.10 | 0.94 | 1.45 | 1.18 | 1.28 | 1.33 | 1.41 |
| Median | 1.21 | 0.90 | 1.05 | 1.36 | 0.94 | 1.55 | 0.75 | 1.34 | 0.66 | 1.26 |
| Maximum | 49.25 | 41.77 | 88.45 | 39.92 | 33.00 | 29.11 | 56.86 | 36.30 | 182.16 | 45.30 |
| Minimum | −21.59 | −44.17 | −52.59 | −35.94 | −18.76 | −31.41 | −36.36 | −21.46 | −67.77 | −29.86 |
| Std. dev. | 6.74 | 8.30 | 13.69 | 8.29 | 6.75 | 7.87 | 8.51 | 6.74 | 12.27 | 8.44 |
| Skewness | 0.76 | −0.12 | 0.56 | 0.20 | 0.26 | −0.14 | 0.41 | 0.35 | 5.96 | 0.08 |
| Kurtosis | 8.89 | 9.87 | 8.26 | 6.00 | 4.35 | 3.89 | 7.96 | 4.99 | 89.80 | 4.53 |
| Sample start | 1973-06 | 1973-03 | 1983-02 | 1982-01 | 1973-03 | 1980-12 | 1973-02 | 1962-02 | 1972-07 | 1972-07 |

| Ticker/Metric | DTE | ED | EIX | ETN | ETR | EXC | F | GD | GDP | GE |
|---|---|---|---|---|---|---|---|---|---|---|
| Mean | 0.95 | 0.98 | 1.10 | 1.96 | 0.98 | 1.04 | 1.17 | 1.44 | 0.94 | 0.90 |
| Median | 0.95 | 0.97 | 1.30 | 1.98 | 0.96 | 1.10 | 0.53 | 1.17 | 0.64 | 0.37 |
| Maximum | 54.18 | 45.00 | 25.92 | 72.89 | 39.24 | 30.69 | 127.38 | 34.00 | 99.77 | 37.20 |
| Minimum | −22.41 | −52.50 | −36.90 | −30.33 | −24.54 | −24.14 | −57.88 | −27.95 | −33.17 | −29.84 |
| Std. dev. | 5.70 | 6.07 | 6.65 | 8.40 | 6.84 | 6.47 | 11.04 | 8.10 | 18.54 | 7.39 |
| Skewness | 1.03 | −0.17 | −0.69 | 0.97 | 0.53 | 0.08 | 2.59 | 0.23 | 2.98 | 0.20 |
| Kurtosis | 13.92 | 16.22 | 6.72 | 12.15 | 6.58 | 4.78 | 33.33 | 4.74 | 18.24 | 5.39 |
| Sample start | 1962-02 | 1962-02 | 1973-06 | 1972-07 | 1972-07 | 1973-06 | 1972-07 | 1977-02 | 2017-01 | 1962-02 |

**Table A2.** *Cont.*

| Ticker/Metric | GIS | GM | GT | HAL | HIG | HON | HSY | IBM | IP | IR |
|---|---|---|---|---|---|---|---|---|---|---|
| Mean | 1.24 | 0.78 | 0.81 | 1.08 | 1.29 | 1.27 | 1.48 | 0.84 | 0.91 | 2.72 |
| Median | 1.12 | 0.33 | 0.37 | 1.24 | 1.39 | 1.40 | 1.11 | 0.36 | 0.50 | 1.85 |
| Maximum | 19.75 | 28.10 | 75.56 | 54.90 | 103.57 | 51.05 | 27.60 | 35.38 | 79.83 | 26.70 |
| Minimum | −24.08 | −31.87 | −41.74 | −59.61 | −74.82 | −38.19 | −24.91 | −24.86 | −37.61 | −24.37 |
| Std. dev. | 5.65 | 9.06 | 10.77 | 10.83 | 12.94 | 8.12 | 6.45 | 6.94 | 8.60 | 10.65 |
| Skewness | 0.08 | 0.18 | 0.60 | −0.15 | 1.43 | 0.06 | 0.20 | 0.19 | 1.04 | 0.12 |
| Kurtosis | 3.99 | 4.26 | 8.13 | 6.24 | 22.84 | 6.97 | 4.85 | 4.83 | 13.67 | 3.31 |
| Sample start | 1980-04 | 2010-12 | 1962-02 | 1972-07 | 1996-01 | 1970-02 | 1983-04 | 1962-02 | 1962-02 | 2017-06 |

| Ticker/Metric | JCPNQ | KHC | KMB | KO | KODK | KR | LMT | MMM | MO | MRK |
|---|---|---|---|---|---|---|---|---|---|---|
| Mean | −0.15 | −0.49 | 1.22 | 1.56 | 9.41 | 2.04 | 1.70 | 0.98 | 1.81 | 1.14 |
| Median | −0.26 | 0.12 | 0.85 | 1.33 | −1.60 | 1.62 | 1.33 | 1.10 | 1.91 | 1.05 |
| Maximum | 52.91 | 24.70 | 33.21 | 33.22 | 879.82 | 214.90 | 48.26 | 25.80 | 195.19 | 31.34 |
| Minimum | −47.83 | −30.94 | −17.10 | −29.55 | −72.63 | −67.84 | −38.81 | −27.83 | −69.65 | −26.62 |
| Std. dev. | 11.83 | 9.02 | 5.86 | 6.38 | 97.98 | 13.48 | 8.89 | 6.06 | 10.71 | 6.89 |
| Skewness | 0.05 | −0.45 | 0.82 | 0.18 | 8.23 | 7.80 | 0.22 | −0.01 | 7.94 | −0.05 |
| Kurtosis | 5.44 | 4.51 | 6.31 | 5.59 | 73.33 | 124.80 | 6.38 | 4.68 | 154.67 | 4.00 |
| Sample start | 1973-03 | 2015-08 | 1980-04 | 1962-02 | 2013-10 | 1977-02 | 1977-02 | 1970-02 | 1962-02 | 1970-02 |

| Ticker/Metric | MRO | MSI | NOC | NSG | OXY | PBI | PCG | PEP | PFE | PG |
|---|---|---|---|---|---|---|---|---|---|---|
| Mean | 0.92 | 1.25 | 1.53 | 1.36 | 0.98 | 1.06 | 0.85 | 1.20 | 1.15 | 1.04 |
| Median | 0.44 | 1.10 | 1.67 | 1.31 | 0.76 | 0.54 | 1.13 | 0.93 | 1.05 | 0.79 |
| Maximum | 86.02 | 30.73 | 33.88 | 25.53 | 72.62 | 73.04 | 45.71 | 36.89 | 39.67 | 24.69 |
| Minimum | −60.08 | −33.49 | −35.56 | −31.52 | −64.63 | −48.66 | −45.26 | −28.41 | −24.01 | −35.42 |
| Std. dev. | 10.55 | 9.56 | 8.30 | 7.55 | 9.63 | 10.06 | 8.45 | 6.34 | 7.01 | 5.48 |
| Skewness | 0.91 | −0.14 | −0.04 | −0.07 | 0.64 | 0.64 | −0.39 | 0.07 | 0.28 | −0.34 |
| Kurtosis | 12.41 | 3.64 | 5.16 | 4.20 | 14.90 | 11.64 | 11.37 | 6.76 | 4.77 | 6.10 |
| Sample start | 1970-02 | 1977-02 | 1982-02 | 1982-07 | 1982-02 | 1972-07 | 1972-07 | 1972-07 | 1972-07 | 1962-02 |

| Ticker/Metric | PPG | PREX | ROK | SEE | SLB | SO | UIS | UNP | XEL | XOM |
|---|---|---|---|---|---|---|---|---|---|---|
| Mean | 1.39 | 0.89 | 1.84 | 1.69 | 0.77 | 1.24 | 0.81 | 1.35 | 0.91 | 0.98 |
| Median | 1.42 | 0.00 | 1.93 | 2.00 | 0.50 | 1.28 | 0.06 | 1.48 | 1.18 | 0.93 |
| Maximum | 26.71 | 52.50 | 160.92 | 149.35 | 39.16 | 22.57 | 130.19 | 34.44 | 42.13 | 22.69 |
| Minimum | −32.32 | −63.83 | −57.90 | −63.97 | −49.47 | −14.26 | −55.92 | −33.43 | −58.50 | −25.13 |
| Std. dev. | 7.29 | 10.88 | 11.83 | 11.41 | 9.28 | 4.99 | 16.10 | 7.49 | 6.07 | 5.31 |
| Skewness | 0.06 | −0.53 | 4.96 | 3.98 | −0.20 | 0.02 | 1.30 | 0.05 | −0.96 | 0.03 |
| Kurtosis | 4.58 | 18.46 | 73.04 | 61.54 | 5.76 | 3.89 | 12.92 | 4.77 | 22.91 | 4.51 |
| Sample start | 1980-04 | 2012-03 | 1982-01 | 1980-04 | 1982-01 | 1982-01 | 1972-08 | 1980-02 | 1973-03 | 1962-02 |

This table reports the descriptive statistics for all available data on survivor stocks. The data are downloaded from Yahoo.com and sorted in alphabetical order.

**Table A3.** Descriptive portfolio statistics for the scientific replication.

| | $RET^{ALL}_{SURVIVOR}$ | S&P 500 | SMB | HML | RMW | CMA | UMD |
|---|---|---|---|---|---|---|---|
| Mean | 1.13 | 0.65 | 0.21 | 0.26 | 0.26 | 0.26 | 0.66 |
| Median | 1.23 | 0.91 | 0.09 | 0.25 | 0.22 | 0.11 | 0.71 |
| Maximum | 18.21 | 16.30 | 18.05 | 12.60 | 13.38 | 9.56 | 18.36 |
| Minimum | −20.41 | −21.76 | −14.86 | −14.11 | −18.48 | −6.86 | −34.39 |
| Std. dev. | 4.38 | 4.27 | 3.02 | 2.87 | 2.17 | 1.99 | 4.19 |
| Skewness | −0.31 | −0.44 | 0.33 | 0.01 | −0.33 | 0.32 | −1.28 |
| Kurtosis | 5.80 | 4.87 | 6.02 | 5.39 | 15.38 | 4.61 | 13.19 |

This table reports the descriptive statistics of the survivor stock portfolio, S&P 500 index, and Fama and French (2018) risk factors. The figures are given in terms of percentages. The sample period is from July 1963 to December 2019.

**Table A4.** Regression estimates for the replicated survivor portfolio using different asset pricing models.

| Alpha | S&P 500 | SMB | HML | RMW | CMA | UMD | $R^2$ |
|---|---|---|---|---|---|---|---|
| 0.49 *** | 0.92 *** | | | | | | 0.83 |
| (7.01) | (56.88) | | | | | | |
| 0.40 *** | 0.96 *** | 0.05 ** | 0.30 *** | | | | 0.86 |
| (6.31) | (64.32) | (2.46) | (13.56) | | | | |
| 0.27 *** | 1.00 *** | 0.12 *** | 0.18 *** | 0.27 *** | 0.24 *** | | 0.88 |
| (4.50) | (67.75) | (5.87) | (6.44) | (9.37) | (5.65) | | |
| 0.34 *** | 0.98 *** | 0.12 *** | 0.13 *** | 0.29 *** | 0.27 *** | −0.10 *** | 0.89 |
| (5.72) | (68.03) | (6.17) | (4.69) | (10.37) | (6.58) | (−6.87) | |

** Statistically significant on a 5% level. *** Statistically significant on a 1% level.

This table reports the results of regressing portfolio $RET^{ALL,excess}_{SURVIVOR,t}$ based on the Yahoo database on the excess returns of the S&P 500 index as well as different asset pricing models. Ordinary *t*-statistics are reported in parentheses. The figures are given in terms of percentages. The sample period is from July 1963 to December 2019.

**Table A5.** Multiple equation model analysis of the replicated survivor portfolio.

| Dependent var. | Alpha | $CRSP^{excess}$ | SMB | HML | RMW | CMA | UMD | $R^2$ |
|---|---|---|---|---|---|---|---|---|
| $RET^{ALL,excess}_{SURVIVOR}$ | 0.09 | 0.98 *** | −0.04 * | 0.15 *** | 0.35 *** | 0.31 *** | −0.12 *** | 0.89 |
| | (1.54) | (67.50) | (−1.81) | (5.45) | (12.32) | (7.47) | (−8.14) | |
| $RET^{excess}_{S\&P500}$ | −0.25 *** | 1.00 *** | −0.16 *** | 0.02 *** | 0.06 *** | 0.04 *** | −0.02 *** | 0.99 |
| | (−15.81) | (262.94) | (−30.46) | (3.08) | (7.89) | (3.48) | (−5.16) | |

* Statistically significant on a 10% level. *** Statistically significant on a 1% level.

This table reports the results of regressing portfolio $RET^{ALL,excess}_{SURVIVOR,t}$ based on the Yahoo database on the excess returns of the S&P 500 index as well as other risk factors in Fama and French's (2018) six-factor model. Ordinary *t*-statistics are reported in parentheses. The figures are given in terms of percentages. The sample period is from July 1963 to December 2019.

**Table A6.** Out-of-sample performance of the replicated survivor portfolio.

| Alpha | Dummy | S&P 500 | SMB | HML | RMW | CMA | UMD | $R^2$ |
|---|---|---|---|---|---|---|---|---|
| 0.56 *** | −0.31 * | 0.93 *** | | | | | | 0.83 |
| (7.05) | (−1.88) | (57.01) | | | | | | |
| 0.40 *** | −0.02 | 0.96 *** | 0.05 ** | 0.30 *** | | | | 0.86 |
| (5.55) | (−0.16) | (64.27) | (2.45) | (13.41) | | | | |
| 0.28 *** | −0.03 | 1.00 *** | 0.12 *** | 0.18 *** | 0.27 *** | 0.24 *** | | 0.88 |
| (4.03) | (−0.21) | (67.70) | (5.85) | (6.38) | (9.37) | (5.65) | | |
| 0.37 *** | −0.12 | 0.98 *** | 0.12 *** | 0.13 *** | 0.29 *** | 0.27 *** | −0.10 *** | 0.89 |
| (5.43) | (−0.92) | (68.02) | (6.12) | (4.54) | (10.39) | (6.59) | (−6.92) | |

* Statistically significant on a 10% level. ** Statistically significant on a 5% level. *** Statistically significant on a 1% level.

This table reports the results of regressing portfolio $RET^{ALL,excess}_{SURVIVOR,t}$ based on the Yahoo database on the excess returns of the S&P 500 index as well as different asset pricing models. The regression models include a dummy variable denoted *d* with a value of 0 in the period from July 1963 to March 2007 and a value of 1 in the period April 2007–December 2019. Ordinary *t*-statistics are reported in parentheses. The figures are given in terms of percentages. The sample period is from July 1963 to December 2019.

**Table A7.** Estimating volatility processes for the replicated survivor portfolio.

|  | $\mu$ | $c$ | $\alpha$ | $\beta$ | $\gamma$ | $v$ |
|---|---|---|---|---|---|---|
| S&P 500 | 0.39 *** | 0.11 | 0.20 *** | 0.90 *** | −0.16 *** | 9.53 |
|  | (2.77) | (1.28) | (2.97) | (28.26) | (−3.99) |  |
| All survivors | 0.76 *** | 0.14 * | 0.14 ** | 0.91 *** | −0.20 *** | 10.69 |
|  | (5.44) | (1.79) | (2.40) | (34.06) | (−5.43) |  |

* Statistically significant on a 10% level. ** Statistically significant on a 5% level. *** Statistically significant on a 1% level.

Here, we use Yahoo data and replicate the portfolio of survivor stocks. This table reports the estimates for the EGARCH model with mean equation:

$$R_{i,t}^{excess} = \mu_i + \epsilon_{i,t}$$

$$\epsilon_{i,t} = \zeta_{i,t}\sigma_{i,t},$$

where $R_{i,t}^{excess}$ is the excess return of at time $t$, $i = \{S\&P\ 500_t,\ all\ survivors_t\}$, $\mu_i$ denotes the intercept term of the mean equation, and $\epsilon_{i,t}$ is the residual term at time $t$. The equation for the variance is:

$$ln(\sigma_{i,t}^2) = c_i + \alpha_i \left| \frac{\epsilon_{i,t-1}}{\sigma_{i,t-1}} \right| + \beta_i ln\left(\sigma_{i,t-1}^2\right) + \gamma_i \frac{\epsilon_{i,t-1}}{\sigma_{i,t-1}},$$

where $\sigma_{i,t}^2$ is the conditional variance at time $t$, and the parameter vector $\boldsymbol{\theta}_{BTC} = (\mu_i, c_i, \alpha_i, \beta_i, \gamma_i)$ is estimated using maximum-likelihood estimation. The models assume that the innovation process follows a fat-tailed $t$-distribution (i.e., $\zeta_{BTC,t}|\boldsymbol{\Omega}_{t-1} \sim t(v)$ with $v$ degrees of freedom). The $z$-statistics are given in parentheses. The sample period is from July 1963 to December 2019.

**Table A8.** Multiple equation model analysis of the replicated survivor portfolio and equal-weighted U.S. equity index.

| Dependent var. | Alpha | $CRSP^{excess}$ | SMB | HML | RMW | CMA | UMD | $R^2$ |
|---|---|---|---|---|---|---|---|---|
| $RET_{SURVIVOR}^{ALL,excess}$ | 0.27 *** | 0.87 *** | −0.80 *** | 0.10 ** | 0.19 *** | 0.05 | 0.00 | 0.75 |
|  | (3.02) | (40.75) | (−20.27) | (2.41) | (4.35) | (0.77) | (0.08) |  |
| $RET_{S\&P500}^{excess}$ | −0.28 *** | 0.90 *** | −0.85 *** | −0.05 * | 0.05 * | −0.11 ** | 0.11 *** | 0.90 |
|  | (−5.05) | (68.29) | (−36.06) | (−1.75) | (1.92) | (−2.78) | (7.77) |  |

* Statistically significant on a 10% level. ** Statistically significant on a 5% level. *** Statistically significant on a 1% level.

This table reports the results of regressing portfolio $RET_{SURVIVOR,t}^{ALL,excess}$ based on the Yahoo database on the excess returns of the S&P 500 index as well as other risk factors in Fama and French's (2018) six-factor model. The factor model specification employs the average excess returns of 49 equal-weighted Fama and French U.S. industrial portfolios as proxy for the market factor. Ordinary $t$-statistics are reported in parentheses. The figures are given in terms of percentages. The sample period is from July 1963 to December 2019.

**Table A9.** Out-of-sample performance of all survivors with robust $t$-statistics.

| Alpha | Dummy | S&P 500 | SMB | HML | RMW | CMA | UMD | $R^2$ |
|---|---|---|---|---|---|---|---|---|
| 0.50 *** | −0.32 ** | 0.86 *** |  |  |  |  |  | 0.84 |
| (5.99) | (−2.47) | (33.76) |  |  |  |  |  |  |
| 0.35 *** | −0.12 | 0.89 *** | 0.08 ** | 0.25 *** |  |  |  | 0.88 |
| (4.96) | (−0.94) | (62.46) | (2.36) | (4.38) |  |  |  |  |
| 0.23 *** | −0.12 | 0.92 *** | 0.15 *** | 0.16 *** | 0.26 *** | 0.20 *** |  | 0.90 |
| (3.30) | (−0.99) | (63.69) | (6.89) | (4.99) | (3.63) | (5.30) |  |  |
| 0.30 *** | −0.19 * | 0.91 *** | 0.15 *** | 0.12 *** | 0.27 *** | 0.22 *** | −0.08 *** | 0.90 |
| (4.20) | (−1.71) | (62.52) | (7.30) | (3.86) | (3.70) | (6.13) | (−2.75) |  |

* Statistically significant on a 10% level. ** Statistically significant on a 5% level. *** Statistically significant on a 1% level.

This table reports the results of regressing portfolio $RET^{ALL,excess}_{SURVIVOR,t}$ on the excess returns of the S&P 500 index as well as different asset pricing models. The regression models include a dummy variable denoted $d$ with a value of 0 in the period from July 1963 to March 2007 and a value of 1 in the period April 2007–December 2019. Robust $t$-statistics using the covariance matrix estimator proposed from Newey and West (1987) with lag order $l = 1$ are reported in parentheses. The figures are given in terms of percentages. The sample period is from July 1963 to December 2019.

## Notes

1. See https://fred.stlouisfed.org/series/DDOM01USA644NWDB (accessed on 14 January 2021).

2. As of February 2019 guidance.

3. The value of a stock's market capitalization traded annually should be at least a quarter million dollars of its shares in each of the previous six months.

4. Additional possible advantages include reduced information asymmetry due to greater scrutiny by investors, increased investor recognition as an industry leader, and a decline in shadow costs. See studies by Denis et al. (2003), Chen et al. (2004), Baran and King (2012), and Chan et al. (2013).

5. For instance, the annualized sample average return varies between 13.63% per annum and 13.75% per annum for the arithmetic return and equal-weighted portfolios, respectively.

6. See https://www.globalpapermoney.com/s-p-releases-list-of-86-companies-in-the-s-p-500-since-1957-cms-1023 (accessed on 31 January 2022).

7. The question arises how does the non-survivorship manifest itself over time? The so-called Lindy law could explain this phenomenon. In this regard, Taleb points out that the Lindy effect (or law) corresponds to situations where the conditional expectation of additional life expectancy increases with time, which requires the survival function of survival time to be that of a power law. A discussion of this issue is provided in Taleb's study "Lindy as a Distance from an Absorbing Barrier", which is available at https://www.academia.edu/44944654 (accessed on 31 January 2022). Future research could elaborate on this issue and model the survival, respectively, non-survival functions for companies in the S&P 500. This issue is, however, beyond the scope of this study and therefore left for future research.

8. As a last resort, the stock name was used to find stock return data in the CRSP database. In this regard, a company could change names or the same company could have different stocks. It is important to note that companies could have similar names and one stock could be changed to another one as successor in the CRSP's dataset. Additionally, ticker symbols for companies can change. Hence, we used the output produced from the CRSP database for tickers associated with corresponding company names. Finally, one stock does not necessarily mean one firm in the CRSP database. For instance, a firm could change its stock to be a different one. Moreover, a stock could also belong to different firms. As an example, firm A spins off into X, Y, and Z different firms. The original stock (in terms of its permno in CRSP) stays with firm X. However, the core business of firm A is actually in firm Z. Now firm Z is assigned a new stock (permno). When we have firm Z's name, and we expand its history, we include the original stock for firm A. In the dataset, we used (to be more inclusive) the stock for firm A in the past as well as firm Z's stock.

9. There are good reasons to use equal-weighted portfolios in the present study. Most importantly, market capitalization as a financial variable is pareto distributed, which means that if value-weighted portfolios were used, a very small number of stocks would receive extraordinarily high weights. Hence, value weighting would distort the overall portfolio return distribution. This distortion occurs when variables deviate from the normal distribution. Our sample stocks share one commonality—namely, survivorship. We are mainly interested in this common link, rather than potential size effects.

10. Because the kurtosis of the regression residuals is 9.87, one could argue that standard $t$-statistics are not valid for making statistical inference. If we assume a $t$-distribution with $v = 4.5$ degrees of freedom, the corresponding kurtosis will be 15, which is much larger than 9.74. Using a 5% significance level, the critical value of this distribution is 2.66. Since the $t$-statistic of 7.01 well exceeds 2.66, we can safely deduce that our statistical inference is valid.

11. As an example, the survivor stock portfolio's loading against the profitability factor exceeds the S&P 500's loading by a factor of 6.5 implying that, on average, survivor stocks are considerably more profitable than the average S&P 500.

12. Peak exposure to the value factor was reached in February 2007 at an economic magnitude of 0.37.

13. Note that excess kurtosis is a stylized fact of financial market data. The use of the Gaussian distribution for modeling the conditional volatilities may lead to misleading results. For this reason, we employ $t$-distributions to model the innovation processes, which explicitly takes into account the fat-tailed data observed here.

14. Importantly, our higher cut-off of 3.39 decreases the likelihood that the performance of the survivor stock portfolio diminished in the ex post March 2007 period. Note that the $t$-statistic of $-2.22$ for our dummy variable in the CAPM model specification indicated a significant structural break on a common 5% level using standard critical values.

[15] As precedent, Alexander and Dimitriu (2005) used Yahoo Finance. Data providers such as CRSP impose relatively high charges for data, whereas Yahoo Finance is freely available, thereby expanding research replicability to a larger audience of scholars. Many universities around the world do not subscribe to CRSP due to costs; in such cases, Yahoo Finance is available.

[16] We found that 7.38% of *RTX* returns exceeded 100% in the sample period from February 1970 to December 2019.

[17] These 14 survivor stocks with the longest available data are: Alcoa Corporation (AA), the Boeing Company (BA), Caterpillar Inc. (CAT), Chevron Corporation (CVX), DTE Energy Company (DTE), Consolidated Edison, Inc. (ED), General Electric Company (GE), Goodyear Tire & Rubber Company (GT), International Business Machines Corporation (IBM), International Paper Company (IP), Coca-Cola Company (KO), Altria Group, Inc. (MO), Procter & Gamble Company (PG), and the Exxon Mobil Corporation (XOM). These companies are very old and were originally established between 1823 and 1925. Ten of these 14 companies were originally founded before 1900.

[18] We would like to thank an anonymous reviewer for suggesting this additional robustness check.

[19] We thank an anonymous reviewer for suggesting this additional robustness check.

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
