# Peer review of "On Survivor Stocks in the S&P 500 Stock Index"

_jrfm, doi:10.3390/jrfm15020095_

Round 1

Reviewer 1 Report

I find the research question addressed in this paper quite interesting. However, I believe the authors can improve on the methodology to make their findings stronger and more robust.

1) I think it will be interesting to see some evidence presented about a portfolio consisting of the firms that did not survive. Presumably, if the survivor portfolio has an abnormal return of 0.3-0.4%/month then the non-survivor portfolio will have an alpha of -0.3 to -0.4%/month.

2) I quite like the matching portfolio construction -- equal-weighted survivor firms with a version of an equal-weighted market portfolio constructed from 10 Fama-French industry portfolios. Perhaps a better construct will obtain using the 30 Fama-French industry portfolios? It is an interesting question and will only make your findings stronger.

3) As a follow-up on my previous point, the CRSP database has an equal-weighted version of their market portfolio (CRSP EW) which will be the ideal way to match the equal-weighted survivor portfolio.

4) It wasn't very clear to me how robust the standard errors of the regressions are. Perhaps the authors can check whether the Newey & West (1987, Econometrica) robust standard errors can be used to see if the empirical findings still hold.

5) Finally, I think readers as well as myself might be curious to find some evidence presented on the risk of non-survivorship over time.

I hope my comments can be of use to the authors to make their manuscript even stronger.

Reviewer 2 Report

The article titled "On survivor stocks in the S&P 500 stock index" examines the characteristics of the survivor stocks in the S&P500 index. The author runs in-sample and out-of-sample analyses and sheds light on the characteristics of the survivor stocks in the index portfolio. The findings consider investment styles and risk characteristics. 

I believe this is a well-executed and interesting study. I have just several comments for the author to consider to potentially improve it further.

1. I recommend reorganizing the introduction a bit. I suggest focusing first on the paper's main findings and, subsequently, discussing the contribution and related literature. The author may try to follow John Cochrane's writing advice, from which I benefitted myself greatly, and I would suggest the author do the same (https://static1.squarespace.com/static/5e6033a4ea02d801f37e15bb/t/5eda74919c44fa5f87452697/1591374993570/phd_paper_writing.pdf).

2. In the data section, it would be interesting to illustrate somehow the relative shares of survivors and non-survivors in the index portfolio.

3. The author may consider explaining the advantage of using the seemingly unrelated regressions instead of the classical approach.

4. The tables and figures are not always self-contained. The author may expand the notes above the table further. Points to consider include indicating whether the numbers are in percentage or not or what type of t-statistic is calculated (HAC?).

Good luck with your work!
